Extended Abstract Track

# Charting Flat Minima Using the Conserved Quantities of Gradient Flow

**Bo Zhao***                                                          bozhao@ucsd.edu
*University of California, San Diego*

**Iordan Ganev***                                                    iganev@cs.ru.nl
*Radboud University*

**Robin Walters**                                          rwalters@northeastern.edu
*Northeastern University*

**Rose Yu**                                                          roseyu@ucsd.edu
*University of California, San Diego*

**Nima Dehmamy**                                          nima.dehmamy@ibm.com
*IBM Research*

**Editors:** Sophia Sanborn, Christian Shewmake, Simone Azeglio, Arianna Di Bernardo, Nina Miolane

## Abstract

Empirical studies have revealed that many minima in the loss landscape of deep learning are connected and reside on a low-loss valley. We present a general framework for finding continuous symmetries in the parameter space, which give rise to the low-loss valleys. We introduce a novel set of nonlinear, data-dependent symmetries for neural networks. We then show that conserved quantities associated with linear symmetries can be used to define coordinates along the minima. The distribution of conserved quantities reveals that using common initialization methods, gradient flow only explores a small part of the global minimum. By relating conserved quantities to convergence rate and sharpness of the minimum, we provide insights on how initialization impacts convergence and generalizability. We also find the nonlinear action to be viable for ensemble building to improve robustness under certain adversarial attacks.

**Keywords:** Symmetry, gradient flow, conserved quantity, Lie group, Lie algebra

## 1. Introduction

Training deep neural networks (DNN) is generally a highly non-convex problem. The loss landscape of DNN is expected to be very rugged, with the number of local minima growing rapidly with model size (Bray and Dean, 2007; Şimşek et al., 2021). However, it has been observed that many DNN loss landscapes contain approximately flat directions along which the loss does not change significantly (Freeman and Bruna, 2017; Garipov et al., 2018). The loss landscape is shaped by the model architecture and the dataset. Hence, a natural question is: *How are parameter space symmetries of the architecture related to flat minima?*

---

\* Equal contribution.

# Extended Abstract Track

Intuitively, it is clear that a symmetry may lead flat minima, as applying a symmetry does not change the loss. However, finding such flat directions is mostly done empirically (Freeman and Bruna, 2017; Garipov et al., 2018; Draxler et al., 2018; Benton et al., 2021; Izmailov et al., 2018). Theoretical results on the existence of such flat minima and how to find them remain sparse. Kunin et al. (2021) shows that some flat directions can be traced back to scaling and shift symmetries of the loss landscape. We discover a set of continuous symmetries which keep nonlinear NN invariant under certain conditions.

We show that coordinates along flat minima can be parametrized using conserved quantities. As in Noether's theorem in physics (Noether, 1918), when a continuous symmetry exists, some quantity will remain unchanged during gradient flow dynamics. We generalize the work in Kunin et al. (2021) and discuss an alternative way to derive such quantities. We also derive the explicit form of conserved quantities for different continuous symmetries of NN. Complete theoretical results appear in Appendix B and C.

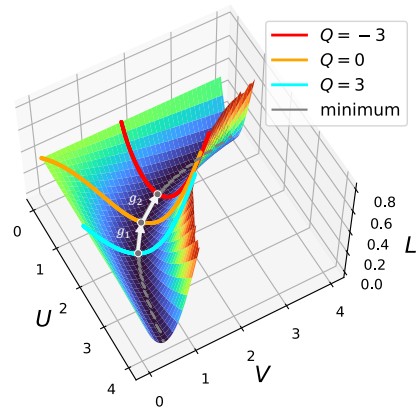

Figure 1: Extended minimum in a 2-layer network $\mathcal{L} = \|Y - UVX\|^2$. Points along the minimum are related by scaling symmetry $U \to Ug^{-1}$ and $V \to gV$. The conserved quantity, $Q = U^2 - V^2$, parametrizes trajectories and minimum.

## 2. Continuous Symmetries in Deep Learning

Let $G$ be a group. An action of $G$ on the parameter space **Param** is a function $\cdot : G \times$ **Param** $\to$ **Param**, written as $g \cdot \boldsymbol{\theta}$, that satisfies the unit and multiplication axioms of the group, meaning $I \cdot \boldsymbol{\theta} = \boldsymbol{\theta}$ where $I$ is the identity of $G$, and $g_1 \cdot (g_2 \cdot \boldsymbol{\theta}) = (g_1 g_2) \cdot \boldsymbol{\theta}$ for all $g_1, g_2 \in G$. The action $G \times$ **Param** $\to$ **Param** is a symmetry of $\mathcal{L}$ if it leaves the loss function invariant, that is:

$$\mathcal{L}(g \cdot \boldsymbol{\theta}) = \mathcal{L}(\boldsymbol{\theta}), \qquad \forall \boldsymbol{\theta} \in \textbf{Param}, \quad g \in G. \tag{1}$$

The groups we discuss below are all matrix Lie groups. Any smooth action of such a group induces an action of the infinitesimal generators of the group, i.e., elements of its Lie algebra. To describe this action, let $\mathfrak{g} = \text{Lie}(G) = T_I G$ be the Lie algebra, which can be thought of as a certain subspace of matrices in $\mathfrak{gl}_n = \mathbb{R}^{n \times n}$, or equivalently, as the tangent space at the identity $I$ of $G$. Given an action of $G$ on the parameter space, we have a vector field for every element $M$ of the Lie algebra $\mathfrak{g}$, known as the *infinitesimal action* of $M$:

$$\overline{M}_{\boldsymbol{\theta}} := \frac{d}{dt}\bigg|_{t \to 0} (\exp_M(t) \cdot \boldsymbol{\theta}) \qquad \forall \boldsymbol{\theta} \in \textbf{Param}. \tag{2}$$

**Proposition 1 (Infinitesimal symmetry)** *Let $G$ be a matrix Lie group acting on parameter space and leaving the loss invariant. Then the gradient vector field is point-wise orthogonal to the infinitesimal action vector field of any Lie algebra element $M \in \mathfrak{g}$:*

$$\langle \nabla_{\boldsymbol{\theta}} \mathcal{L} \, , \, \overline{M}_{\boldsymbol{\theta}} \rangle = 0 \qquad \forall \boldsymbol{\theta} \in \textbf{Param}. \tag{3}$$

In the special case where $G$ acts linearly on parameter space, the infinitesimal action of the matrix $M \in \mathfrak{g}$ on **Param** is defined in terms of matrix multiplication: $\overline{M}_{\boldsymbol{\theta}} = M \cdot \boldsymbol{\theta}$. Examples of linear symmetries in homogeneous and radial neural networks have been discussed in literatures, which we summarize in Appendix B.3. Below we show a more interesting application of Proposition 1 in nonlinear neural networks.

**Proposition 2 (Nonlinear symmetries)** *Consider the $F(X) = U\sigma(VX)$ network for a fixed batch $X \in \mathbb{R}^{m \times k}$. Let $G \subseteq \mathrm{GL}_h(\mathbb{R})$ be such that $\forall g \in G$, $\sigma(gVX)$ is full rank (rank $\min(h, k)$. Consider the following nonlinear action of $G$ on* **Param** $\times$ **Data**

$$g \cdot (U, V, X) = (U\pi(g, VX),\ gV,\ X) \qquad\qquad \pi(g, Z) = \sigma(Z)\sigma(gZ)^+ \qquad (4)$$

*where $\pi : G \times \mathbb{R}^{h \times k} \to \mathrm{GL}_h(\mathbb{R})$ is a nonlinear representation of $G$ and $A^+$ is the the pseudoinverse of $A$. Then, the action (4) is a symmetry of* **Param** *which keeps $F(X)$ invariant if $h = k$.*

This result applies a pair of consecutive layers in arbitrary architectures and can easily generalize to multilayer neural networks. One consequence of these continuous symmetries is the existence of extended, flat minima (Appendix B.6). Next, we discuss how these symmetries give rise to conserved quantities during gradient flow (GF) dynamics.

## 3. Conserved Quantities of Gradient Flow

Gradient descent can be regarded as a discrete approximation of a continuous gradient flow process, where $t \in \mathbb{R}_{>0}$. In gradient flow, the parameters have the dynamics

$$\dot{\boldsymbol{\theta}}(t) = d\boldsymbol{\theta}(t)/dt = -\varepsilon\nabla\mathcal{L}(\boldsymbol{\theta}(t)). \qquad (5)$$

A *conserved quantity* of GF is a function $Q : $ **Param** $\to \mathbb{R}$ such that for any two time points $s, t \in \mathbb{R}_{>0}$ along a GF trajectory $Q(\boldsymbol{\theta}(s)) = Q(\boldsymbol{\theta}(t))$. In other words, we have $dQ(\boldsymbol{\theta}(t))/dt = 0$. A few conserved $Q$ of GF have been discovered, most notably layer imbalance in homogeneous feed-forward networks (Du et al., 2018). We generalize these results to the symmetries discussed above.

**Proposition 3** *Suppose the action of $G$ on* **Param** *is linear and leaves $\mathcal{L}$ invariant. For any $M \in \mathfrak{g}$, the function $Q_M : $ **Param** $\to \mathbb{R}$ is a conserved quantity:*

$$Q_M(\boldsymbol{\theta}) = \langle \boldsymbol{\theta}, M \cdot \boldsymbol{\theta} \rangle \qquad (6)$$

*The space of distinct conserved quantities of the form $Q_M$ for $M \in \mathfrak{g}$ is in one-to-one correspondence with the space of symmetric matrices in $\mathfrak{g}$.*

The conserved $Q_M$ are only nonzero for the symmetric part of $M$. For all $M$, the flow lines satisfy the differential equation $\dot{\boldsymbol{\theta}}^T M \boldsymbol{\theta} = 0$, which has the form of an angular momentum when $M$ is anti-symmetric. More details and examples are in Appendix C.3.

In physics, Noether's theorem (Noether, 1918) states that continuous symmetries give rise to conserved quantities. Recently, Tanaka and Kunin (2021) showed that Noether's

# Extended Abstract Track

theorem can also be applied to gradient descent by approximating it as a *second order* GF. In the limit where the second order GF reduces to first order GF (23), results from Noether's theorem reduce to our conservation law $\langle \overline{M}_{\boldsymbol{\theta}}, \nabla \mathcal{L} \rangle = 0$ (11) in Appendix C.4.

The conserved quantities we derived so far assumes continuous gradient flow. In gradient descent, the values of these conserved quantities may change due to the time discretization. However, the change in $Q$ is expected to be small. For example, in two-layer linear networks, the change of $Q$ is bounded by the square of learning rate. Appendix D contains derivations and empirical observations of the magnitude of change in $Q$.

## 4. Applications and Future Work

**Relation to convergence rate and generalizability** Conserved quantities are determined at initialization and by definition unchanged during gradient flows. By relating the values of conserved quantities to optimization and model generalizability, we know some properties of the trajectory and the final model before the gradient flow even starts. This knowledge allows us to choose good conserved quantity values at initialization, in order to influence the convergence rate or the sharpness of the trained model. In Appendix F, we derive the relation between conserved quantities and convergence rate for two optimization examples. We show that initializing parameters with certain conserved quantity values accelerates convergence. In Appendix G, we derive the relation between conserved quantities and sharpness of solutions in a simple two-layer network, and show empirically that conserved quantity values affect sharpness in larger networks.

**Ensemble models** While the set of possible values for a conserved quantity is often unbounded, common initialization methods such as Xavier initialization (Glorot and Bengio, 2010) limit the values of $Q$ to a small range (Appendix E). As a result, only a small part of the minimum is reachable by the models. Applying the group actions allow us to obtain an ensemble without any retraining or searching. We show that even with stochasticity in the data, the loss is approximately unchanged under the group action. The ensemble has the potential to improve robustness under adversarial attacks (Appendix H).

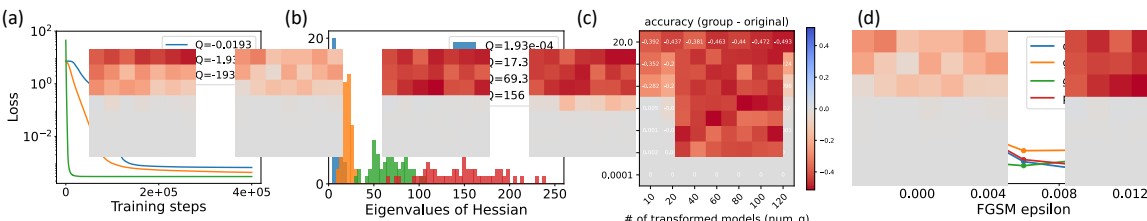

Figure 2: Overview of empirical observations with more details in Appendix F, G, and H. (a) In a two-layer neural network, the convergence rate depends on the conserved quantity $Q$. (b) The distribution of the eigenvalues of the Hessian at the minimum is related to $Q$. (c) The ensemble created by group actions has similar loss values when $\varepsilon$ is small. (d) The ensemble model improves robustness against fast gradient sign method attacks.

## Acknowledgments

This work was supported in part by U.S. Department Of Energy, Office of Science grant DE-SC0022255, U. S. Army Research Office grant W911NF-20-1-0334, and NSF grants #2134274 and #2146343. I. Ganev was supported by the NWO under the CORTEX project (NWA.1160.18.316). R. Walters was supported by the Roux Institute and the Harold Alfond Foundation and NSF grants #2107256 and #2134178.

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

# Extended Abstract Track

Pierre Foret, Ariel Kleiner, Hossein Mobahi, and Behnam Neyshabur. Sharpness-aware minimization for efficiently improving generalization. In *International Conference on Learning Representations*, 2020.

C Daniel Freeman and Joan Bruna. Topology and geometry of half-rectified network optimization. In *5th International Conference on Learning Representations, ICLR 2017*, 2017.

Iordan Ganev, Twan van Laarhoven, and Robin Walters. Universal approximation and model compression for radial neural networks. *arXiv preprint arXiv:2107.02550v2*, 2022.

Timur Garipov, Pavel Izmailov, Dmitrii Podoprikhin, Dmitry P Vetrov, and Andrew G Wilson. Loss surfaces, mode connectivity, and fast ensembling of dnns. *Advances in neural information processing systems*, 31, 2018.

Xavier Glorot and Yoshua Bengio. Understanding the difficulty of training deep feedforward neural networks. In *Proceedings of the thirteenth international conference on artificial intelligence and statistics*, pages 249–256. JMLR Workshop and Conference Proceedings, 2010.

Grzegorz Głuch and Rüdiger Urbanke. Noether: The more things change, the more stay the same. *arXiv preprint arXiv:2104.05508*, 2021.

Sepp Hochreiter and Jürgen Schmidhuber. Flat minima. *Neural computation*, 9(1):1–42, 1997.

Dongsung Huh. Curvature-corrected learning dynamics in deep neural networks. In *International Conference on Machine Learning*, pages 4552–4560. PMLR, 2020.

Pavel Izmailov, Dmitrii Podoprikhin, Timur Garipov, Dmitry Vetrov, and Andrew Gordon Wilson. Averaging weights leads to wider optima and better generalization. *arXiv preprint arXiv:1803.05407*, 2018.

Nitish Shirish Keskar, Dheevatsa Mudigere, Jorge Nocedal, Mikhail Smelyanskiy, and Ping Tak Peter Tang. On large-batch training for deep learning: Generalization gap and sharp minima. *International Conference on Learning Representations*, 2017.

Minyoung Kim, Da Li, Shell X Hu, and Timothy Hospedales. Fisher sam: Information geometry and sharpness aware minimisation. In *International Conference on Machine Learning*, pages 11148–11161. PMLR, 2022.

Daniel Kunin, Javier Sagastuy-Brena, Surya Ganguli, Daniel LK Yamins, and Hidenori Tanaka. Neural mechanics: Symmetry and broken conservation laws in deep learning dynamics. In *International Conference on Learning Representations*, 2021.

Hancheng Min, Salma Tarmoun, René Vidal, and Enrique Mallada. On the explicit role of initialization on the convergence and implicit bias of overparametrized linear networks. In *International Conference on Machine Learning*. PMLR, 2021.

Emmy Noether. Invariante variationsprobleme. *Nachrichten von der Gesellschaft der Wissenschaften zu Göttingen, Mathematisch-Physikalische Klasse*, page 235–257, 1918.

Henning Petzka, Michael Kamp, Linara Adilova, Cristian Sminchisescu, and Mario Boley. Relative flatness and generalization. *35th Conference on Neural Information Processing Systems*, 2021.

Andrew M. Saxe, James L. McClelland, and Surya Ganguli. Exact solutions to the nonlinear dynamics of learning in deep linear neural networks. *arXiv preprint arXiv:1312.6120v3*, 2014.

Berfin Şimşek, François Ged, Arthur Jacot, Francesco Spadaro, Clément Hongler, Wulfram Gerstner, and Johanni Brea. Geometry of the loss landscape in overparameterized neural networks: Symmetries and invariances. In *International Conference on Machine Learning*, pages 9722–9732. PMLR, 2021.

Hidenori Tanaka and Daniel Kunin. Noether's learning dynamics: Role of symmetry breaking in neural networks. *Advances in Neural Information Processing Systems*, 34, 2021.

Salma Tarmoun, Guilherme Franca, Benjamin D Haeffele, and Rene Vidal. Understanding the dynamics of gradient flow in overparameterized linear models. In *International Conference on Machine Learning*, pages 10153–10161. PMLR, 2021.

Andre Wibisono and Ashia C Wilson. On accelerated methods in optimization. *arXiv preprint arXiv:1509.03616*, 2015.

## Appendix A. Related Work

**Continuous symmetry in parameter space**   Overparametrization in neural networks leads to symmetries in the parameter space Głuch and Urbanke (2021). Continuous symmetry has been identified in fully-connected linear networks (Tarmoun et al., 2021), homogeneous neural networks (Badrinarayanan et al., 2015; Du et al., 2018), radial neural networks (Ganev et al., 2022), and softmax and batchnorm functions (Kunin et al., 2021).

**Conserved quantities**   The imbalance between layers in linear or homogeneous networks is known to be invariant during gradient flow and related to convergence rate (Saxe et al., 2014; Du et al., 2018; Arora et al., 2018a,b; Tarmoun et al., 2021; Min et al., 2021). Huh (2020) discovered similar conservation laws in natural gradient descents. Kunin et al. (2021) develop a more general approach for finding conserved quantities for certain one-parameter symmetry groups. In physics, Noether's theorem gives a conserved quantity for every continuous symmetry Noether (1918). However, while symmetry of the loss function may be known, the kinetic energy makes it difficult to find symmetry of the Lagrangian that describes the learning dynamics. Tanaka and Kunin (2021) takes a different route by studying the dynamics of the conserved quantity caused by this kinetic symmetry breaking.

**Topology of minimum** The global minimum of overparametrized neural networks are connected spaces instead of isolated points. Cooper (2018) proves that the global minima is usually a manifold with dimension equal to the number of parameters subtracted by the number of data points. Şimşek et al. (2021) study permutation symmetry and show that in certain overparametrized networks, the minimum related by permutations are connected. Entezari et al. (2021) hypothesize that SGD solutions can be permuted to points on the same connected minima. Ainsworth et al. (2022) develop algorithms that find such permutations.

## Appendix B. Continuous symmetries in deep learning

In this section, we first summarize our notation for basic neural network constructions. Then we consider group actions on the parameter space that leave the loss invariant. We prove that, in the case of such an action, the gradient vector field is orthogonal to the infinitesimal action vector field corresponding to any Lie algebra element.

### B.1. The parameter space and loss function

The parameters of a neural network consist of a choice of weight matrix[1] for each layer. In symbols, one has $W_i \in \mathbb{R}^{n_i \times m_i}$ for each layer $i$, where $n_i$ and $m_i$ are the layer output and input dimensions, respectively. For feedforward networks, successive input and output dimensions match: $m_i = n_{i-1}$. We group the widths into a tuple $\mathbf{n} = (n_L, \dots, n_1, n_0)$, and the parameter space becomes[2]: $\mathbf{Param}(\mathbf{n}) = \mathbb{R}^{n_L \times n_{i-1}} \times \dots \times \mathbb{R}^{n_1 \times n_0}$. We denote an element therein as a tuple of matrices $\boldsymbol{\theta} = (W_i \in \mathbb{R}^{n_i \times n_{i-1}})_{i=1}^{L}$. The activation of the $i$-th layer is a piecewise differentiable function $\sigma_i : \mathbb{R}^{n_i} \to \mathbb{R}^{n_i}$, which can be pointwise, as is conventionally the case, but is not necessarily so.

Fix parameter values $\boldsymbol{\theta} \in \mathbf{Param}$. For any input vector $x \in \mathbb{R}^{n_0}$, we set $H_{\boldsymbol{\theta},i}(x)$, or simply $H_i(x)$, to be the feature vector in the $i$-th layer, before the activation is applied. Hence we have the recursion $H_{i+1} = W_{i+1} \star \sigma(H_i)$, where $\star$ denotes the matrix product, convolution, or other ways that the weights may act on the input features. For simplicity, we largely focus on the case of multilayer perceptrons (MLPs), when $\star$ denotes matrix multiplication. The feedforward function $F_{\boldsymbol{\theta}} : \mathbb{R}^{n_0} \to \mathbb{R}^{n_L}$ corresponding to parameters $\boldsymbol{\theta} \in \mathbf{Param}$ is then defined as $F_{\boldsymbol{\theta}}(x) = \sigma_L(H_{\boldsymbol{\theta},L}(x))$.

The "loss function" $\mathcal{L}$ of our model is defined as:

$$\mathcal{L} : \mathbf{Param} \times \mathbf{Data} \to \mathbb{R}, \qquad \mathcal{L}(\boldsymbol{\theta}, (x, y)) = \mathrm{Cost}(y, F_{\boldsymbol{\theta}}(x)). \tag{7}$$

where $\mathbf{Data} = \mathbb{R}^{n_0} \times \mathbb{R}^{n_L}$ is the space of possible training data pairs, and $\mathrm{Cost} : \mathbb{R}^{n_L} \times \mathbb{R}^{n_L} \to \mathbb{R}$ is a differentiable cost function, such as mean square error or cross-entropy. This loss function generalizes to multiple samples, where the data consists of matrices $X \in \mathbb{R}^{n_0 \times k}$ and $Y \in \mathbb{R}^{n_L \times k}$ whose columns are the $k$ samples. (We use capital letters for matrix data and small letters for individual samples, but retain the same notation for the feedforward function, i.e., $F_{\boldsymbol{\theta}} : \mathbb{R}^{n_0 \times k} \to \mathbb{R}^{n_L \times k}$.) In the results appearing below, we prove properties of $\mathcal{L}$ that hold for any training data. Hence, unless specified otherwise, we take a fixed batch of training data $\{(x_i, y_i)\}_{i=1}^{k} \subseteq \mathbf{Data}$, and consider the loss to be a function of the parameters only.

---

1. For the purposes of exposition, we suppress the bias vectors.
2. When clear from context, we omit $\mathbf{n}$ and write just $\mathbf{Param}$.

# Extended Abstract Track

**Example 1 (Two-layer network with MSE)** *Consider a network with* $\mathbf{n} = (n, h, m)$, *the identity output activation, and no biases. The parameter space is* $\mathbf{Param}(\mathbf{n}) = \mathbb{R}^{n \times h} \times \mathbb{R}^{h \times m}$ *and we denote an element as* $\boldsymbol{\theta} = (U, V)$. *Taking the mean square error cost function, the loss function for data* $(X, Y) \in \mathbb{R}^{n \times k} \times \mathbb{R}^{m \times k}$ *takes the form* $\mathcal{L}(\boldsymbol{\theta}, (X, Y)) = \frac{1}{k} \| Y - U\sigma_1(VX)\|^2$.

## B.2. Action of continuous groups and infinitesimal symmetries

Let $G$ be a group. An action of $G$ on the parameter space $\mathbf{Param}$ is a function $\cdot : G \times \mathbf{Param} \to \mathbf{Param}$, written as $g \cdot \boldsymbol{\theta}$, that satisfies the unit and multiplication axioms of the group, meaning $I \cdot \boldsymbol{\theta} = \boldsymbol{\theta}$ where $I$ is the identity of $G$, and $g_1 \cdot (g_2 \cdot \boldsymbol{\theta}) = (g_1 g_2) \cdot \boldsymbol{\theta}$ for all $g_1, g_2 \in G$ .

**Definition 4 (Parameter space symmetry)** *We say that the action* $G \times \mathbf{Param} \to \mathbf{Param}$ *is a symmetry of* $\mathcal{L}$ *if it leaves the loss function invariant, that is:*

$$\mathcal{L}(g \cdot \boldsymbol{\theta}) = \mathcal{L}(\boldsymbol{\theta}), \qquad \forall \boldsymbol{\theta} \in \mathbf{Param}, \quad g \in G \qquad (8)$$

The groups we discuss below are all matrix Lie groups, that is, continuous subgroups $G \subseteq \mathrm{GL}_n(\mathbb{R})$ of the general linear group of invertible $n \times n$ real matrices, for some $n$. Any smooth action of such a group induces an action of the infinitesimal generators of the group, i.e., elements of its Lie algebra. To describe this action, let $\mathfrak{g} = \mathrm{Lie}(G) = T_I G$ be the Lie algebra, which can be thought of as a certain subspace of matrices in $\mathfrak{gl}_n = \mathbb{R}^{n \times n}$, or (equivalently) as the tangent space at the identity $I$ of $G$. With the former realization in mind, for every matrix $M \in \mathfrak{g}$, we have an exponential map $\exp_M : \mathbb{R} \to G$ valued in the group, defined as $\exp_M(t) = \sum_{k=0}^{\infty} \frac{(tM)^k}{k!}$. Given an action of $G$ on the parameter space, we have a vector field for every element $M$ of the Lie algebra $\mathfrak{g}$, known as the *infinitesimal action* of $M$:

$$\text{Infinitesimal action of } M \text{ vector field:} \qquad \overline{M}_{\boldsymbol{\theta}} := \frac{d}{dt}\bigg|_{t \to 0} (\exp_M(t) \cdot \boldsymbol{\theta}) \qquad \forall \boldsymbol{\theta} \in \mathbf{Param}. \qquad (9)$$

In the case of a parameter space symmetry, the invariance of $\mathcal{L}$ translates into the following orthogonality condition, where the inner product $\langle , \rangle : \mathbf{Param} \times \mathbf{Param} \to \mathbb{R}$ is calculated by contracting all indices, e.g. $\langle A, B \rangle = \sum_{ijk...} A_{ijk...} B_{ijk...}$.

**Proposition 5 (Infinitesimal symmetry)** *Let $G$ be a matrix Lie group acting parameter space and leaving the loss function invariant. Then the gradient vector field is point-wise orthogonal to the infinitesimal action vector field of any Lie algebra element* $M \in \mathfrak{g}$:

$$\langle \nabla_{\boldsymbol{\theta}} \mathcal{L} , \overline{M}_{\boldsymbol{\theta}} \rangle = 0 \qquad \forall \boldsymbol{\theta} \in \mathbf{Param} \qquad (10)$$

**Proof** The gradient is the transpose of the Jacobian, so the left-hand-side becomes $d\mathcal{L}_{\boldsymbol{\theta}} \left( \frac{d}{dt}\big|_0 (\exp_M(t) \cdot \boldsymbol{\theta}) \right)$, which is equal to $\frac{d}{dt}\big|_0 \mathcal{L}(\exp_M(t) \cdot \boldsymbol{\theta})$ by the chain rule. The invariance of $\mathcal{L}$ implies that $\mathcal{L}(\exp_M(t) \cdot \boldsymbol{\theta}) = \mathcal{L}(\boldsymbol{\theta})$ for any $t$. The result follows. ∎

# Extended Abstract Track

In the special case where $G$ acts linearly on parameter space, we have the following reformulation. The general linear group GL(**Param**) consists of all invertible linear transformations[3] of the parameter space. Suppose $G$ is a subgroup of GL(**Param**), with the resulting linear action on **Param**. Then its Lie algebra $\mathfrak{g}$ is a Lie subalgebra of $\mathfrak{gl}_d = \mathbb{R}^{d \times d}$ where $d = \dim(\mathbf{Param})$, and the infinitesimal action of the matrix $M \in \mathfrak{g}$ on **Param** is defined in terms of matrix multiplication: $\overline{M}_{\boldsymbol{\theta}} = M \cdot \boldsymbol{\theta}$. A corollary of Proposition 5 is:

**Corollary 6** *Suppose $G$ acts linearly on the parameter space leaving the loss function invariant. Then the gradient of the loss function and the multiplication action of any $M \in \mathfrak{g}$ are orthogonal:*

$$\langle \nabla_{\boldsymbol{\theta}} \mathcal{L} \, , \, M \cdot \boldsymbol{\theta} \rangle = 0 \qquad \forall \boldsymbol{\theta} \in \mathbf{Param} \tag{11}$$

## B.3. Examples of linear symmetries

We now describe a linear group action on the parameter space with a cancellation property on successive layers. Such an action exists for a vast swath of different architectures. Before discussing the general case, however, we first analyze two-layer networks, hence illustrating the main ideas in as simple a way as possible. To this end, consider a two-layer network with dimension vector $(m, h, n)$, so that the parameter space consists of pairs of matrices $(U, V) \in \mathbb{R}^{m \times h} \times \mathbb{R}^{h \times n}$. Let $\sigma = \sigma_1$ be the hidden layer activation and assume (for simplicity) that there is no output activation. The feedforward function is therefore $F(x) = U\sigma(Vx)$ for $x \in \mathbb{R}^n$. Let $G \subseteq \mathrm{GL}_h(\mathbb{R})$ be a subgroup, and let $\pi : G \to \mathrm{GL}_h(\mathbb{R})$ a representation; the simplest example is when $\pi(g) = g$. Define the action of $G$ on **Param** as follows:

$$g \cdot U = U\pi(g^{-1}), \qquad g \cdot V = gV \tag{12}$$

**Proposition 7 (Equivariant activation)** *Suppose $\sigma(gz) = \pi(g)\sigma(z)$ for all $z \in \mathbb{R}^h$. Then the action given in 12 is a symmetry of the parameter space.*

**Proof** The loss depends on the parameters only through the feedforward function, so it suffices to show that $(U, V)$ and $g \cdot (U, V)$ give the same feedforward function. The key computation is: $(g \cdot U)\sigma((g \cdot V)x) = U\pi(g^{-1})\sigma(gVx) = U\sigma(Vx)$, where $x \in \mathbb{R}^n$ is any input vector. ∎

To describe the infinitesimal action, note that the representation $\pi : G \to \mathrm{GL}_h$ induces a representation $d\pi : \mathfrak{g} \to \mathfrak{gl}_h$ of the Lie algebra, defined by $d\pi(M) = \frac{d}{dt}\big|_0 \pi(\exp_M(t))$. Using the fact that $\exp_{-M}(t) = \exp_M(t)^{-1}$, the infinitesimal action of the Lie algebra $\mathfrak{g}$ induced by 12 is given by:

$$M \cdot U = -U d\pi(M), \qquad M \cdot V = MV \tag{13}$$

The infinitesimal version of $\sigma(gz) = \pi(g)\sigma(z)$ is

$$\langle Mz, d\sigma_z \rangle = d\pi(M)\sigma(z) \tag{14}$$

---

3. Concretely, let $d = \dim(\mathbf{Param})$ be the dimension of the parameter space; in terms of the widths, we have $d = \sum_{i=1}^L n_i n_{i-1}$. Then $\mathbf{Param} \simeq \mathbb{R}^d$ and parameters can be 'flattened' into $d$-vectors. One can identify GL(**Param**) with the group $\mathrm{GL}_d(\mathbb{R})$ of invertible $d \times d$ matrices, and its Lie algebra with $\mathfrak{gl}_d = \mathbb{R}^{d \times d}$.

where $d\sigma_z \in \mathbb{R}^{h \times h}$ is the Jacobian matrix of $\sigma : \mathbb{R}^h \to \mathbb{R}^h$ at $z \in \mathbb{R}^h$. When the activation is elementwise, we have $Mz \odot \sigma'(z) = d\pi(M)\sigma(z)$ ($\odot$ denotes elementwise multiplication) [4]

**Example 2 (Linear networks)** *The simplest example of Proposition 7 is that of linear networks, where $\sigma(x) = x$. One can take $\pi(g) = g$ and $G = \mathrm{GL}_h(\mathbb{R})$.*

**Example 3 (Homogeneous activations)** *Suppose the activation $\sigma : \mathbb{R}^h \to \mathbb{R}^h$ is homogeneous, so that $\sigma$ is applied pointwise in the standard basis, and there exists $\alpha > 0$ such that $\sigma(cz) = c^\alpha \sigma(z)$ for all $c \in \mathbb{R}_{>0}$ and $z \in \mathbb{R}^h$. Define the positive scaling group $G$ to be the subgroup of $\mathrm{GL}_h$ consisting of diagonal matrices with positive diagonal entries. We show that, for any $g \in G$, and any $z \in \mathbb{R}^h$, we have $\sigma(gz) = g^\alpha \sigma(z)$. Indeed, let $g$ be an element of the positive rescaling group, so $g = \mathrm{diag}(\mathbf{c})$ is a diagonal matrix, where the diagonal entries $\mathbf{c} = (c_1, \ldots, c_h)$ satisfy $c_i \in \mathbb{R}_{>0}$. For $z = (z_1, \ldots, z_h) \in \mathbb{R}^h$, we have: $\sigma(gz) = \sum_j \sigma(c_j z_j) = \sum_j c_j^\alpha \sigma(z_j) = g^\alpha \sigma(z)$, as desired. Hence, we set $\pi(g) = g^\alpha$. We have $d\pi(M) = \alpha M$ and so the infinitesimal version of rescaling invariance of homogeneous $\sigma$ becomes $Mz \odot \sigma'(z) = \alpha M \sigma(z)$.*

**Example 4 (LeakyReLU)** *A special type of homoegeneous activation is LeakyReLU (including ReLU), defined as $\sigma(z) = \max(z, 0) - s\min(z, 0)$, where $s$ is positive real number. We have that $\alpha = 1$, and take $\pi(g) = g$. Since $\sigma(z) = z\sigma'(z)$, infinitesimal equivariance becomes $Mz \odot \sigma'(z) = M\sigma(z)$.*

**Example 5 (Radial rescaling activations)** *A less trivial example of continuous symmetries is the case of a radial rescaling activation (Ganev et al., 2022) where for $x \in \mathbb{R}^h \backslash \{0\}$, $\sigma(x) = f(\|x\|)x$ for some function $f$. Radial rescaling activations are is equivariant under rotations of the input: for any orthogonal transformation $g \in O(h)$ (that is, $g^T g = I$) we have $\sigma(gz) = g\sigma(z)$ for all $z \in \mathbb{R}^h$. Indeed, $\sigma(gz) = f(\|gz\|)(gz) = g(f(\|z\|)z) = g\sigma(z)$, where we use the fact that $\|gz\| = z^T g^T gz = z^T z = \|z\|$ for $g \in O(h)$. Hence, Proposition 7 is satisfied with $\pi(g) = g$.*

### B.4. Nonlinear Symmetries

So far we assumed the group action $g \cdot \boldsymbol{\theta}$ is linear. We now introduce a set of nonlinear group actions that can keep the the loss invariant. Consider the equivariance $\pi(g)\sigma(z) = \sigma(gz)$ from Prop. 7. Consider our example subnetwork $F(z) = U\sigma(Vz)$ with dimensions $(n, h, m)$. Most $\sigma$ may not be equivariant under the linear action of the full $\mathrm{GL}_h(\mathbb{R})$ group. However, if we generalize Prop. 7 and allow $\pi$ to act nonlinearly, it is possible to make many $\sigma$ equivariant under the full $\mathrm{GL}_h(\mathbb{R})$.

Let $H_b^a$ denote entry in row $a$ and column $b$ of a batch of inputs $X \in \mathbb{R}^{m \times k}$. Assume $\sigma$ acts independently on each sample $a$ (column), $\sigma(X)_a = \sigma(X_a)$. We can define the following nonlinear symmetry for a fixed input:

**Proposition 8 (Nonlinear symmetries)** *Consider the $F(X) = U\sigma(VX)$ network for a fixed batch $X \in \mathbb{R}^{m \times k}$. Let $G \subseteq \mathrm{GL}_h(\mathbb{R})$ be such that $\forall g \in G$, $\sigma(gVX)$ is full rank (rank*

---

4. The general case is $d\sigma_z \cdot (Mz) = d\pi(M) \cdot \sigma(z)$, where $d\sigma_z \in \mathbb{R}^{h \times h}$ is the Jacobian matrix of $\sigma : \mathbb{R}^h \to \mathbb{R}^h$ at $z \in \mathbb{R}^h$, and $\cdot$ denotes matrix multiplication.

$\min(h, k)$. *Consider the following nonlinear action of $G$ on* **Param** $\times$ **Data**

$$g \cdot (U, V, X) = (U\pi(g, VX), \ gV, \ X) \qquad\qquad \pi(g, Z) = \sigma(Z)\sigma(gZ)^+ \qquad (15)$$

*where* $\pi : G \times \mathbb{R}^{h \times k} \to \mathrm{GL}_h(\mathbb{R})$ *is a nonlinear representation of $G$ and $A^+$ is the the pseudoinverse of $A$. Then, the action* (15) *is a symmetry of* **Param** *which keeps $F(X)$ invariant if $h = k$.*

**Proof** We can check that the group axioms are satisfied when $k \leq h$. For brevity, let $Z = VX$ so that $g \cdot (U, Z) = (U\pi(g, Z), gZ)$. Setting $g = I$, we see that the identity action holds when $\sigma(Z)^+\sigma(Z) = I_k$, which requires $k \leq h$. For the group multiplication, for $g_1, g_2 \in G$ we have

$$g_2 \cdot g_1 \cdot (U, Z) = g_2 \cdot (U\pi(g_1, Z), g_1 Z)$$
$$= (U\pi(g_1, Z)\pi(g_2, g_1 Z), g_2 g_1 Z)$$
$$\pi(g_1, Z)\pi(g_2, g_1 Z) = \sigma(Z)\sigma(g_1 Z)^+\sigma(g_1 Z)\sigma(g_2 g_1 Z)^+$$
$$= \sigma(Z)\sigma(g_2 g_1 Z)^+ = \pi(g_2 g_1, Z)$$
$$\Rightarrow g_2 \cdot g_1 \cdot (U, Z) = [g_2 g_1] \cdot (U, Z) \qquad (16)$$

Setting $g_2 = g_1^{-1}$, (16) shows the inverse axiom also holds. Lastly, in order for this action to be a symmetry, the identity action should yield $I \cdot (U, V, X) = (U, V, X)$, meaning $\pi(I, Z) = \sigma(Z)\sigma(Z)^+ = I_h$. This only holds if $h \geq k$. As a result, (15) is a symmetry if $h = k$. ∎

Note that the nonlinear symmetry above is restricted to a given input $X \in \mathbb{R}^{h \times h}$ (setting $h = h$ to have a symmetry). This means that a transformed set of parameters $(U', V')$ defined by $(U', V', X) = g \cdot (U, V, X)$ will not yield the same output as $(U, V)$ when used on a different input $X^*$ because $\pi(g^{-1}, VX)\sigma(gVX*) \neq \sigma(VX^*)$. We cannot use it to define transformed parameters $(U', V') = g(U, V)$ one batch $X$ to transform parameters to $(U', V') = (U\pi(g, VX), gV)$, while $U'\sigma(V'X) = U\sigma(VX)$, for different batch $X' \in \mathbb{R}^{h \times h}$ using $\pi(g, VX)$ may not keep $F(VX)$

### B.5. Linear symmetries: multilayer case

We now state a more general version of the results of the previous section by combining equivariances across all layers of a multilayer network. Specifically, consider a feedforward fully-connected neural network with widths $\mathbf{n} = (n_0, \ldots, n_L)$, so that the parameters space consists of tuples of matrices $\boldsymbol{\theta} = (W_i \in \mathbb{R}^{n_i \times n_{i-1}})_{i=1}^L$. For each layer $1 < i < L$, let $G_i$ be a subgroup of $\mathrm{GL}_{n_i}$, and let $\pi_i : G_i \to \mathrm{GL}_{n_i}(\mathbb{R})$ be a representation (in many cases, we take $\pi_i(g) = g$). Define an action of $G = G_1 \times \cdots \times G_L$ on **Param** via

$$\forall g = (g_1, \ldots, g_L) \in G, \qquad\qquad g \cdot W_i = g_{i+1} W_i \pi_i(g_i^{-1}) \qquad (17)$$

For each $i$, the representation $\pi_i$ induces a Lie algebra representation $d(\pi_i) : \mathfrak{g}_i \to \mathfrak{gl}_{n_i}$, as described above. The infinitesimal action of the Lie algebra $\mathfrak{g} = \mathfrak{g}_1 \times \cdots \times \mathfrak{g}_L$ induced by 17 is given by:

$$\forall M = (M_1, \ldots, M_L) \in \mathfrak{g}, \qquad\qquad g \cdot M_i = M_{i+1} W_i - W_i d(\pi_i)(M_i) \qquad (18)$$

The proof of the first part of the following Proposition proceeds by induction, where the key computation is that of from Proposition 7. The second part relies on Corollary 6.

Extended Abstract Track

**Proposition 9** *Suppose that, for each $i = 1, \ldots, L$, the activation $\sigma_i$ intertwines the two actions of $G_i$, that is, $\sigma_i(g_i z_i) = \pi_i(g_i)\sigma(z_i)$ for all $g_i \in G_i$, $z_i \in \mathbb{R}^{n_i}$. Then:*

1. *(Combined equivariance of activations) The action of $G = G_1 \times \cdots \times G_L$ defined in 17 is a symmetry of the parameter space.*

2. *(Infinitesimal equivariant action) The action of $\mathfrak{g} = \mathfrak{g}_1 \times \cdots \times \mathfrak{g}_L$ defined in 18 satisfies $\langle \nabla_\theta \mathcal{L}, M \cdot \theta \rangle$ for all $\theta \in$ **Param** and all $M \in \mathfrak{g}$.*

Next, we discuss the effects of continuous symmetry on the loss landscape.

### B.6. Linear symmetries lead to extended, flat minima

In this section, we show that, in the case of a linear group action, applying the action of any element of the group to a local minimum yields another local minimum. This fact is a corollary of a more general result; in order to describe it and remove ambiguity, we include the following clarifications. Let $G$ be a matrix Lie group acting as a linear symmetry. Fix a basis $(\theta_1, \ldots, \theta_d)$ of the parameter space. The gradient $\nabla_\theta \mathcal{L}$ of the loss $\mathcal{L}$ at a point $\theta \in$ **Param** in the parameter space as another vector in **Param** $\simeq \mathbb{R}^d$, whose $i$-th coordinate is the partial derivative $\frac{\partial \mathcal{L}}{\partial \theta_i}\big|_\theta$. Hence, it makes sense to apply the group action to the gradient: $g \cdot \nabla_\theta \mathcal{L}$. We regard vectors in **Param** $\simeq \mathbb{R}^d$ as column vectors with $d$ rows. Thus, the transpose of any vector is a row vector with $d$ columns. In the case of the gradient, its transpose at $\theta$ matches the *Jacobian* $d\mathcal{L}_\theta \in \mathbb{R}^{1 \times d}$ of $\mathcal{L}$, that is: $d\mathcal{L}_\theta = (\nabla_\theta \mathcal{L})^T$. Alternative notation for the Jacobian is $d\mathcal{L}_{\theta_0} = \frac{\partial \mathcal{L}}{\partial \theta}\big|_{\theta_0}$, where we now use $\theta$ as a dummy variable and $\theta_0 \in$ **Param** as a specific value. As noted above, we are interested in matrix Lie groups $G \subseteq \mathrm{GL}_d(\mathbb{R}) = \mathrm{GL}(\mathbf{Param})$, and assume that the matrix transpose $g^T$ belongs to $G$ for any $g \in G$. These assumptions hold in all examples of interest.

**Proposition 10** *Suppose the action of $G$ on the parameter space is linear and leaves the loss invariant. Then the gradients of $\mathcal{L}$ at any $\theta_0$ and $g \cdot \theta_0$ are related as follows:*

$$g^T \cdot \nabla_{g \cdot \theta_0} \mathcal{L} = \nabla_{\theta_0} \mathcal{L} \qquad \forall g \in G, \ \forall \theta_0 \in \mathbf{Param} \tag{19}$$

**Proof** Let $T_g : \mathbf{Param} \to \mathbf{Param}$ be the transformation corresponding to $g \in G$. The the Jacobian $d\mathcal{L}_{\theta_0}$ is given by:

$$d\mathcal{L}_{\theta_0} = \frac{\partial \mathcal{L}}{\partial \theta}\bigg|_{\theta_0} = \frac{\partial (\mathcal{L} \circ T_g)}{\partial \theta}\bigg|_{\theta_0} = \frac{\partial \mathcal{L}}{\partial \theta}\bigg|_{g \cdot \theta_0} \frac{\partial T_g}{\partial \theta}\bigg|_{\theta_0} = d\mathcal{L}_{g \cdot \theta_0} \circ T_g$$

where we have used the definition of the Jacobian, the invariance of the loss ($\mathcal{L} \circ T_g = \mathcal{L}$), the chain rule, the linearity of the action. The result follows from applying $T_{g^{-1}}$ on the right to both sides, and using the fact that the gradient is the transpose of the Jacobian. ∎

Now suppose $\theta^*$ is a critical point of $\mathcal{L}$ so that $\nabla_{\theta^*} \mathcal{L} = 0$. Then Proposition 10 implies that $g \cdot \theta^*$ is another critical point: $\nabla_{g \cdot \theta^*} = (g^T)^{-1} \nabla_{\theta^*} \mathcal{L} = 0$. If the group $G$ is connected (as the case in most applications), then continuity implies that critical points that are local minima are sent to other local minimum. We summarize this discussion in the following corollary:

# Extended Abstract Track

**Corollary 11** *Suppose $G$ is connected. If $\boldsymbol{\theta}^*$ is a critical point (resp. local minimum) of $\mathcal{L}$, then so is $g \cdot \boldsymbol{\theta}^*$.*

Hence, if $\boldsymbol{\theta}^*$ is a critical point, then the set $\{g \cdot \boldsymbol{\theta} \mid g \in G\}$ belongs to the critical locus. This set is known as the *orbit* of $\boldsymbol{\theta}$ under the action of $G$, and is isomorphic to the quotient $G/\mathrm{Stab}_G(\boldsymbol{\theta})$, where $\mathrm{Stab}_G(\boldsymbol{\theta}) = \{g \in G \mid g \cdot \boldsymbol{\theta} = \boldsymbol{\theta}\}$ is the *stabilizer* subgroup of $\boldsymbol{\theta}$ in $G$. In the case of a linear action, the orbit is a smooth manifold. While the corollary above implies that the critical locus is a union of $G$-orbits, it does not imply, in general, that the critical locus is a single $G$-orbit. It also does not rule out the case that the stabilizer is a somewhat 'large' subgroup of $G$, in which case the orbit would have low dimension.

**Example 6 (Running example)** *Recall that the parameter space of a two-layer network is $\mathbf{Param} = \mathbb{R}^{m \times h} \times \mathbb{R}^{h \times n}$, where the dimension vector is $(m, h, n)$, and we write elements as $(U, V)$. Pick such an element in which $U$ and $V$ have full rank (this is a generic condition).*

- **Linear case**. *The $\mathrm{GL}_h$-orbit has dimension $h^2 - \max(0, h - n)\max(0, h - m)$.*

- **Homogeneous case**. *The orbit of the positive rescaling group has dimension $\min(h, \max(n, m))$.*

- **Radial rescaling case**. *The orbit of the orthogonal group has dimension $\binom{h}{2}$ if $h \leq \max(n, m)$ and dimension $\binom{h}{2} - \binom{h - \max(m,n)}{2}$ otherwise.*

*To summarize:*

$$
\begin{array}{lll}
\text{Linear case} & \dim = h^2 - \max(0, h - n)\max(0, h - m) & (20) \\[1em]
\text{Homogeneous case} & \dim = \min(h, \max(n, m)) & (21) \\[1em]
\text{Radial rescaling case} & \dim = \begin{cases} \binom{h}{2} & \text{if } h \leq \max(n, m) \\ \binom{h}{2} - \binom{h - \max(m,n)}{2} & \text{otherwise} \end{cases} & (22)
\end{array}
$$

Next, we discuss how these symmetries give rise to conserved quantities during "gradient flow" (GF) dynamics (continuous version of gradient descent (GD)).

## Appendix C. Conserved quantities of gradient flow

Thus far, we discussed a class of continuous symmetries of the loss in neural networks and how it may lead to extended loss minima, which we will call "valleys". Symmetries can move us along a loss valley. The questions that arises are: *How do we distinguish between different minima along a valley? Can we parametrize points along a loss valley? Does training using SGD typically end up in a specific part of these valleys?* We will show now that these questions can be addressed using conserved quantities.

In brief, we show that certain quantities remain constant during gradient flow (GF) (continuous version of gradient descent (GD)). We will denote these conserved quantities by $Q$. The value of $Q$ is fixed by the initialization and remains constant during GF. As we show, common initialization schemes, such as Xavier, lead to a narrow distribution of values for $Q$. As a result, if stochasticity is not too high, SGD will only converge to and explore a small portion of a loss valley. Our goal is to provide a systematic way to explore the rest of the valley. By applying symmetries we can change the value of $Q$ and thereby explore parts of loss valleys which SGD seldom converges to.

# Extended Abstract Track

## C.1. Gradient flow (GF)

The update rule for GD is $\boldsymbol{\theta}_{t+1} = \boldsymbol{\theta}_t - \varepsilon \nabla \mathcal{L}(\boldsymbol{\theta}_t)$ where $\varepsilon$ is the learning rate, which in general can be a symmetric matrix. GD can be regarded as a discrete approximation of a continuous GF process, where $t \in \mathbb{R}_{>0}$. In GF, the parameters have the dynamics

$$\dot{\boldsymbol{\theta}}(t) = d\boldsymbol{\theta}(t)/dt = -\varepsilon \nabla \mathcal{L}(\boldsymbol{\theta}(t)). \tag{23}$$

At time $t = 0$, initializing at $\boldsymbol{\theta}(0) \in \textbf{Param}$, GF defines a "trajectory" $\boldsymbol{\theta}(t) \in \textbf{Param}$ for $t \in \mathbb{R}_{>0}$.

## C.2. Conserved quantities

A *conserved quantity* of GF is a function $Q : \textbf{Param} \to \mathbb{R}$ such that for any two time points $s, t \in \mathbb{R}_{>0}$ along a GF trajectory $Q(\boldsymbol{\theta}(s)) = Q(\boldsymbol{\theta}(t))$. In other words, we have $dQ(\boldsymbol{\theta}(t))/dt = 0$. A few conserved $Q$ of GF have been discovered, most notably layer imbalance $Q_{\text{imb}} \equiv \|W_i\|^2 - \|W_{i-1}\|^2$ (Du et al., 2018) for each pair of feedforward linear ($\sigma(x) = x$) layers $i - 1, i$, and its full matrix version $Q_i = W_i^T W_i - W_{i-1} W_{i-1}^T$. It is known that imbalance and certain other conserved $Q$ can be related to symmetries (Huh, 2020; Kunin et al., 2021). We generalize these results to the symmetries discussed above. Using chain rule and the GF equation we have

$$\frac{dQ}{dt} = \left\langle \frac{\partial Q}{\partial \boldsymbol{\theta}} \bigg|_{\boldsymbol{\theta}(t)}, \frac{d\boldsymbol{\theta}(t)}{dt} \right\rangle = -\langle \nabla Q, \varepsilon \nabla \mathcal{L} \rangle = 0 \tag{24}$$

which also equals $-\langle \varepsilon \nabla Q, \nabla \mathcal{L} \rangle = 0$ because $\varepsilon = \varepsilon^T$. Note that, if $f : \mathbb{R} \to \mathbb{R}$ is any function, and $Q$ is a conserved quantity, the $f \circ Q$ is also a conserved quantity. Additionally, any linear combination of conserved quantities is again a conserved quantity.

## C.3. Conserved quantities from symmetries

For simplicity, set the learning rate to identity, $\varepsilon = I$. Notice the similarity of $dQ/dt = 0$ (24) with infinitesimal group action (11) $\langle M \cdot \boldsymbol{\theta}, \nabla \mathcal{L} \rangle = 0$. We can associate a conserved $Q$ to a symmetry by finding solutions to $\nabla Q(\boldsymbol{\theta}) = M \cdot \boldsymbol{\theta}$ (or more generally to $f(\boldsymbol{\theta}) \nabla Q(\boldsymbol{\theta}) = M \cdot \boldsymbol{\theta}$, with $f : \textbf{Param} \to \mathbb{R}$). Alternatively, we can use the GF equation to write (11) as $\dot{Q}_M \equiv \left\langle \dot{\boldsymbol{\theta}}, M \cdot \boldsymbol{\theta}, \right\rangle = 0$. Matrices in the Lie algebra $\mathfrak{g}$ give rise to conserved quantities for the gradient flow, as explained in the following proposition:

**Proposition 12** *Suppose the action of $G$ on* **Param** *is linear[5] and leaves $\mathcal{L}$ invariant. For any $M \in \mathfrak{g}$, the function $Q_M :$ **Param** $\to \mathbb{R}$ is a conserved quantity:*

$$Q_M(\boldsymbol{\theta}) = \langle \boldsymbol{\theta}, M \cdot \boldsymbol{\theta} \rangle \tag{25}$$

*The space of distinct conserved quantities of the form $Q_M$ for $M \in \mathfrak{g}$ is in one-to-one correspondence with the space of symmetric matrices in $\mathfrak{g}$.*

---

5. For simplicity, we also assume that $G$ is closed under taking transposes, and that $G$ acts *faithfully* . These assumptions generally hold in practice.

**Proof** Flattening $\boldsymbol{\theta}$ into a $d = \prod \mathbf{n}$ dimensional vector, we get $M \in \mathbb{R}^{d \times d}$. Assuming, $M^T \in \mathfrak{g}$, let $M^{\mathrm{sym}} = \frac{1}{2}(M + M^T) \in \mathfrak{g}$. Using GF $\dot{\theta} = -\nabla \mathcal{L}$, we have $Q_M = \boldsymbol{\theta}^T M \boldsymbol{\theta} = Q_{M^{\mathrm{sym}}}$ and $\dot{Q}_M = \left\langle \dot{\boldsymbol{\theta}}, M \cdot \boldsymbol{\theta} \right\rangle + \left\langle \boldsymbol{\theta}, M \cdot \dot{\boldsymbol{\theta}} \right\rangle = -\left\langle \nabla \mathcal{L}, M^{\mathrm{sym}} \cdot \boldsymbol{\theta} \right\rangle = 0$ ∎

The conserved $Q_M = Q_{M_S}$ above are only nonzero for the symmetric part of $M$. If $M$ is an anti-symmetric matrix in $\mathfrak{g}$, then $Q_M \equiv 0$, so we do not obtain meaningful conserved quantities. Instead, we can only conclude that flow lines satisfy the differential equation $\dot{\boldsymbol{\theta}}^T M \boldsymbol{\theta} = 0$. Selecting coordinates $e_1, \dots, e_d$ for the parameter space **Param**, this equation becomes:

$$\sum_{i < j} M_{ij} r_{ij}^2 \dot{\phi}_{ij} \equiv 0 \tag{26}$$

where $(r_{ij}, \phi_{ij})$ are the polar coordinates for the point $(\boldsymbol{\theta}_i, \boldsymbol{\theta}_j) \in \mathbb{R}^2$. In summary, we find:

| $M \in \mathfrak{g}$ | $M$ symmetric | $M$ anti-symmetric |
|---|---|---|
| differential equation | conserved quantity | differential equation |
| $\dot{\theta}^T M \theta = 0$ | $Q_M(\theta) = \theta^T M \theta$ | $\sum_{i<j} m_{ij} r_{ij}^2 \dot{\phi}_{ij} \equiv 0$ |

**Example 7 (Imbalance in linear layers)** *Let $\sigma_i(z) = z$ in an $L$ layer feedforward NN ($i < L$). $\mathcal{L}$ is invariant under $G_i = \mathrm{GL}_{n_i}(\mathbb{R})$. Let $M \in \mathfrak{gl}_{n_i}$ be symmetric. The corresponding $Q_M$ is given by*

$$Q_M = \mathrm{Tr}\left[ W_i^T M_i W_i \right] - \mathrm{Tr}\left[ W_{i+1} M W_{i+1}^T \right] \tag{27}$$

*Using the one-hot matrices $E_{kl}$ (only component $k, l$ is 1, rest are 0) and setting $M = E_{\{kl\}} = E_{kl} + E_{lk}$, we get $Q_M = \left[ W_i W_i^T - W_{i+1}^T W_{i+1} \right]_{kl}$. This shows that each component $kl$ of the whole matrix $\mathbf{Q} = W_i W_i^T - W_{i+1}^T W_{i+1}$ is conserved in linear layers.*

**Example 8 (Equivariant $\sigma$)** *Let $\sigma$ be equivariant under a linear action of $g \in G \subseteq \mathrm{GL}_h(\mathbb{R})$ with $\pi(g)\sigma(z) = \sigma(gz)$. Let $F(z) = U\sigma(Vz)$ be two layers in an NN with loss invariant under the above $G$. For symmetric $M \in \mathfrak{g}$, and using the infinitesimal action (13), the following $Q_M$ is conserved*

$$Q_M = \mathrm{Tr}\left[ U d\pi(M) U^T \right] - \mathrm{Tr}\left[ V^T M V \right]. \tag{28}$$

**Example 9 (Homogeneous activation under scaling)** *For $g \in G = \mathbb{R}_{>0}^h$ (positive scaling group) we have $\sigma(gz) = g^\alpha \sigma(z)$ and $d\pi(M) = \alpha M$, which yields the conserved $Q_M = \alpha \mathrm{Tr}\left[ U M U^T \right] - \mathrm{Tr}\left[ V^T M V \right]$. Note, for scaling all $M$ are diagonal and hence symmetric. Using the basis $M = E_{kk}$, we see that the set of all conserved $Q_M$ becomes $\mathbf{Q} = \mathrm{diag}\left[ \alpha U^T U - V V^T \right]$ (here, $\mathrm{diag}[A]$ is the leading diagonal). Special cases of this are LeakyReLU and ReLU with $\alpha = 1$.*

**Example 10 (Radial rescaling activation)** *Consider the $F(z) = U\sigma(Vz)$ network with dimensions $(n, h, m)$ and let $\sigma(z) = f(\|z\|)z$ be the radial rescaling activation. As in sec. B.3, $G = O(h)$ commutes with $\sigma$, making it a symmetry of $\mathcal{L}$. The Lie algebra $\mathfrak{g}$ consists*

*only of anti-symmetric matrices $M = -M^T$, and so from (25) yields $Q_M = 0$ and no nontrivial $Q_M$ can be found this way. However, $M \in \mathfrak{g}$ still satisfies*

$$\boldsymbol{\theta}^T M \dot{\boldsymbol{\theta}} = -\operatorname{Tr}\left[UM\dot{U}^T\right] + \operatorname{Tr}\left[V^T M \dot{V}\right] = 0, \tag{29}$$

*and (26). Using the canonical basis of $E_{[kl]} \in \mathfrak{g} = o(h)$ given by $E_{[kl]} = E_{kl} - E_{lk}$ ([kl] meaning anti-symmetrized indices), (29) becomes*

$$\boldsymbol{\theta}^T E_{[kl]} \dot{\boldsymbol{\theta}} = [\dot{U}^T U]_{[kl]} - [\dot{V}V^T]_{[kl]} = \sum_{s=1}^{n} r_{U,s}^2 \dot{\phi}_{U,s} - \sum_{s=1}^{m} r_{V^T,s}^2 \dot{\phi}_{V^T,s} = 0 \tag{30}$$

*$(r_{U,s}, \phi_{U,s})$ and $(r_{V^T,s}, \phi_{V^T,s})$ are the polar coordinates of row $U_s$ and column $V^s = V_s^T$ in the 2D plane with basis $(e^k, e^l)$ [6]. This is analogous to the "angular momentum" in 2D $x \wedge \dot{x} = r^2 \dot{\phi}$. Intuitively, (30) states that in every 2D plane $(k, l)$, the angular momenta of rows of $U$ and columns of $V$ are equal. This result trivially generalizes to multi-layer networks, with $U = W_i$ and $V = W_{i-1}$.*

### C.4. Relation to Noether's theorem

We now show how the approach in (Tanaka and Kunin, 2021) relates to our conservation law $dQ/dt = \left\langle \dot{\boldsymbol{\theta}}, M\boldsymbol{\theta} \right\rangle = 0$. Assuming a small time-step $\tau \ll 1$, we can write GD as $\boldsymbol{\theta}(t + \tau) - \boldsymbol{\theta}(t) = -\tilde{\varepsilon}\nabla\mathcal{L}(\boldsymbol{\theta}(t))$. Expanding the l.h.s to second order in $\tau$ and discarding $O(\tau^3)$ terms defines the 2nd order GF equation

$$\text{2nd order GF:} \qquad \frac{d\boldsymbol{\theta}}{dt} + \frac{\tau}{2}\frac{d^2\boldsymbol{\theta}}{dt^2} = -\varepsilon\nabla\mathcal{L}. \tag{31}$$

Here $\varepsilon = \tilde{\varepsilon}/\tau$. To use Noether's theorem, the dynamics (i.e. GF) must be a variational (Euler-Lagrange (EL)) equation derived from an "action" $S(\boldsymbol{\theta})$ (objective function), which for (31) is the time integral of Bregman Lagrangian $L$ (Wibisono and Wilson, 2015)

$$S(\boldsymbol{\theta}) = \int dt L(\boldsymbol{\theta}(t), \dot{\boldsymbol{\theta}}(t); t) = \int_{\gamma} dt e^{t/\tau}\left[\frac{\tau}{2}\left\langle \dot{\boldsymbol{\theta}}, \varepsilon^{-1}\dot{\boldsymbol{\theta}}\right\rangle - \mathcal{L}(\boldsymbol{\theta})\right] \tag{32}$$

where $\boldsymbol{\theta} : \mathbb{R} \to \textbf{Param}$ is a trajectory (flow path) in **Param**, parametrized by $t$. The variational EL equations find the paths $\gamma^*$ which minimize the action, meaning $\partial S_{\gamma}/\partial \gamma|_{\gamma^*} = 0$.

**Noether's theorem** states that if $M \in \mathfrak{g}$ is a symmetry of the *action* $S(\boldsymbol{\theta})$ (32) (not just the loss $\mathcal{L}(\boldsymbol{\theta})$), then the Noether current $J_M$ is conserved

$$\text{Noether current:} \quad J_M = \left\langle \overline{M}\boldsymbol{\theta}, \frac{\partial L}{\partial \dot{\boldsymbol{\theta}}} \right\rangle = e^{t/\tau}\left\langle \overline{M}\boldsymbol{\theta}, \varepsilon^{-1}\dot{\boldsymbol{\theta}}\right\rangle = e^{t/\tau}J_{0M},$$

$$\text{Conservation:} \quad \frac{dJ_M}{dt} = e^{t/\tau}\left[\frac{1}{\tau}J_{0M} + \frac{dJ_{0M}}{dt}\right] = 0, \quad \Rightarrow \quad J_{0M}(t) = J_{0M}e^{-t/\tau} \tag{33}$$

---

6. The polar coordinate are found by setting $U_{sl} = r_{U,s}\cos\phi_{U,s}$ and $U_{sk} = r_{U,s}\sin\phi_{U,s}$ (same for $V^T$).

Tanaka and Kunin (2021) also derived the Noether current (33), but concludes that because $L(\boldsymbol{\theta}, \dot{\boldsymbol{\theta}}) \neq \mathcal{L}(\boldsymbol{\theta})$, the symmetries are "broken" and therefore doesn't derive conserved charges for the types of symmetries we discussed above. However, while Tanaka and Kunin (2021) focuses on 2nd order GF, we note that our conserved $Q$ were derived for *first order GF*, which is found from the $\tau \to 0$ limit of 2nd oder GF. In this limit $L \to e^{t/\tau}\mathcal{L}$ and thus symmetries of $L$ also becomes symmetries of $\mathcal{L}$. When $\tau \to 0$, 2nd order GF reduces to $\varepsilon^{-1}\dot{\boldsymbol{\theta}} = -\nabla\mathcal{L}$ the conserved charge goes to

$$\lim_{\tau \to 0} J_{0M} = \left\langle \overline{M}\boldsymbol{\theta}, \nabla\mathcal{L} \right\rangle = J_{0M}(0) \lim_{\tau \to 0} e^{-t/\tau} = 0, \tag{34}$$

which means that we recover the invariance under infinitesimal action (11). In fact, for linear symmetries and symmetric $M \in \mathfrak{g}$, $J_{0M} = dQ_M/dt = 0$.

## Appendix D. Gradient descent and drifting conserved quantities

Due to the non-infinitesimal time steps, conserved quantities in gradient flow are no longer conserved in gradient descent. However, with small learning rate, we expect the change in the conserved quantities to be small. In this section, we first prove that the change of $Q$ is bounded by the square of learning rate for two layer linear networks, then show empirically that the change $Q$ is small for nonlinear networks.

### D.1. Change in $Q$ in gradient descent (linear layers)

**Proposition 13** *Consider the two layer linear network, where $U \in \mathbb{R}^{m \times h}, V \in \mathbb{R}^{h \times n}$ are the only parameters, and the loss function $\mathcal{L}$ is a function of $UV$. In gradient descent with learning rate $\eta$, the change in the conserved quantity $Q = \mathrm{Tr}\left[U^T U - VV^T\right]$ at step $t$ is bounded by*

$$|Q_{t+1} - Q_t| \leq \eta^2 \left|\frac{d\mathcal{L}(t)}{dt}\right|. \tag{35}$$

**Proof** Let $U_t$ and $V_t$ be the value of $U$ and $V$ at time $t$ in a gradient descent. The update rule is

$$U_{t+1} = U_t - \eta\frac{\partial\mathcal{L}}{\partial U}, \qquad\qquad V_{t+1} = V_t - \eta\frac{\partial\mathcal{L}}{\partial V} \tag{36}$$

Consider the two layer linear reparametrization $W = UV$.

$$Q_t = \mathrm{Tr}\left[U_t^T U_t - V_t V_t^T\right]$$

$$Q_{t+1} = \mathrm{Tr}\left[U_{t+1}^T U_{t+1} - V_{t+1}V_{t+1}^T\right]$$

$$= \mathrm{Tr}\left[\left(U_t - \eta\frac{\partial\mathcal{L}(U_t)}{\partial U_t}\right)^T \left(U_t - \eta\frac{\partial\mathcal{L}(U_t)}{\partial U_t}\right) - \left(V_t - \eta\frac{\partial\mathcal{L}(V_t)}{\partial V_t}\right)\left(V_t - \eta\frac{\partial\mathcal{L}(V_t)}{\partial V_t}\right)^T\right] \tag{37}$$

Expanding $Q_{t+1}$ and subtracting by $Q_t$, we have

$$Q_{t+1} - Q_t = \mathrm{Tr}\left[\eta^2\left(\frac{\partial\mathcal{L}}{\partial U_t}\right)^T \frac{\partial\mathcal{L}}{\partial U_t} - \eta\left(\frac{\partial\mathcal{L}}{\partial U_t}\right)^T U_t - \eta U_t^T\frac{\partial\mathcal{L}}{\partial U_t} - \eta^2\frac{\partial\mathcal{L}}{\partial V_t}\left(\frac{\partial\mathcal{L}}{\partial V_t}\right)^T + \eta\frac{\partial\mathcal{L}}{\partial V_t}V_t^T + \eta V_t\left(\frac{\partial\mathcal{L}}{\partial V_t}\right)^T\right]$$

$$\tag{38}$$

Note that

$$\left(\frac{\partial \mathcal{L}}{\partial U_t}\right)^T U_t = (\nabla \mathcal{L} V_t^T)^T U_t = V_t \nabla \mathcal{L}^T U_t = V_t \left(\frac{\partial \mathcal{L}}{\partial V_t}\right)^T, \tag{39}$$

and similarly

$$U_t^T \frac{\partial \mathcal{L}}{\partial U_t} = \frac{\partial \mathcal{L}}{\partial V_t} V_t^T. \tag{40}$$

Therefore, 38 simplifies to

$$Q_{t+1} - Q_t = \eta^2 \operatorname{Tr}\left[\left(\frac{\partial \mathcal{L}}{\partial U_t}\right)^T \frac{\partial \mathcal{L}}{\partial U_t} - \frac{\partial \mathcal{L}}{\partial V_t}\left(\frac{\partial \mathcal{L}}{\partial V_t}\right)^T\right]$$

$$= \eta^2 \left(\operatorname{Tr}\left[\left(\frac{\partial \mathcal{L}}{\partial U_t}\right)^T \frac{\partial \mathcal{L}}{\partial U_t}\right] - \operatorname{Tr}\left[\left(\frac{\partial \mathcal{L}}{\partial V_t}\right)^T \frac{\partial \mathcal{L}}{\partial V_t}\right]\right), \tag{41}$$

and the variation of Q in each step is bounded by the convergence rate:

$$|Q_{t+1} - Q_t| = \eta^2 \left|\operatorname{Tr}\left[\left(\frac{\partial \mathcal{L}}{\partial U_t}\right)^T \frac{\partial \mathcal{L}}{\partial U_t}\right] - \operatorname{Tr}\left[\left(\frac{\partial \mathcal{L}}{\partial V_t}\right)^T \frac{\partial \mathcal{L}}{\partial V_t}\right]\right|$$

$$\leq \eta^2 \left|\operatorname{Tr}\left[\left(\frac{\partial \mathcal{L}}{\partial U_t}\right)^T \frac{\partial \mathcal{L}}{\partial U_t}\right] + \operatorname{Tr}\left[\left(\frac{\partial \mathcal{L}}{\partial V_t}\right)^T \frac{\partial \mathcal{L}}{\partial V_t}\right]\right|$$

$$= \eta^2 \left|\frac{d\mathcal{L}}{dt}\right| \tag{42}$$

$$\blacksquare$$

### D.2. Empirical observations

In gradient flow, the conserved quantity $Q$ is constant by definition. In gradient descent, $Q$ varies with time. In order to see how applicable our theoretical results are in gradient descent, we investigate the amount of variation in $Q$ in gradient descent using two-layer neural networks.

Since $Q$ is the difference between the two terms $f_1(U) = \frac{1}{2}\operatorname{Tr}[U^T U]$ and $f_2(V) = \sum_{a,j}\int_{x_0}^{V_{aj}} dx \frac{\sigma(x)}{\sigma'(x)}$, we normalize $Q$ by the initial value of $f_1(U)$ and $f_2(V)$, i.e.,

$$\tilde{Q} = \frac{\left|\frac{1}{2}\operatorname{Tr}[U^T U] - \sum_{a,j}\int_{x_0}^{V_{aj}} dx \frac{\sigma(x)}{\sigma'(x)}\right|}{\left|\frac{1}{2}\operatorname{Tr}[U_0^T U_0]\right| + \left|\sum_{a,j}\int_{x_0}^{V_{0_{aj}}} dx \frac{\sigma(x)}{\sigma'(x)}\right|}$$

and denote the amount of change in $\tilde{Q}$ as

$$\Delta\tilde{Q}(t) = \tilde{Q}(t) - \tilde{Q}(0) \tag{43}$$

# Extended Abstract Track

We run gradient descent on two-layer networks with whitened input with the following objective

$$\text{argmin}_{U,V}\{\mathcal{L}(U,V) = \|Y - U\sigma(V^T)\|_F^2\} \tag{44}$$

where $\sigma$ is the identity function, ReLU, sigmoid, or tanh. $Y \in \mathbb{R}^{5\times 10}$, $U \in \mathbb{R}^{5\times 50}$ and $V \in \mathbb{R}^{10\times 50}$ have random Gaussian initialization with zero mean. We repeat the gradient descent with learning rate 0.1, 0.01, and 0.001.

The variation $\Delta\tilde{Q}(t)$ and loss is shown in Fig.3. The amount of change in $Q$ is small relative to the magnitude of $f_1(U)$ and $f_2(V)$, indicating that conserved quantities in gradient flow are approximately conserved in gradient descent. The error in $Q$ grows with step size, as $\Delta\tilde{Q}(t)$ is larger with the largest learning rate we used, although it has the same magnitude as those of smaller learning rates. We also observe that $Q$ stays constant after loss converges.

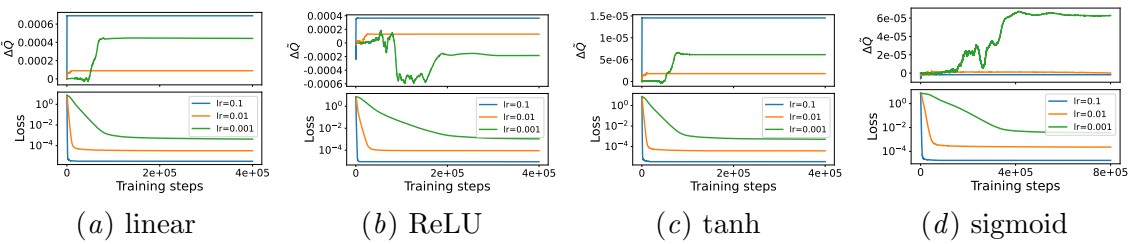

(a) linear  (b) ReLU  (c) tanh  (d) sigmoid

Figure 3: Dynamics of conserved quantities in GD. The amount of change in $Q$ is small relative to its magnitude, and $Q$ converges when loss converges.

## Appendix E. Distribution of $Q$ under Xavier initialization

We first consider a linear two-layer neural network $UVX$, where $U \in \mathbb{R}^{m\times h}, V \in \mathbb{R}^{h\times n}$, and $X \in \mathbb{R}^{n\times k}$. We choose the following form of the conserved quantity:

$$Q = \frac{1}{2}\text{Tr}[U^T U - VV^T]. \tag{45}$$

Xavier initialization keeps the variance of each layer's output the same as the variance of the input. Under Xavier initialization (Glorot and Bengio, 2010), each element in a given layer is initialized independently, with mean 0 and variance equal to the inverse of the layer's input dimension:

$$U_{ij} = \mathcal{N}\left(0, \frac{1}{h}\right) \qquad\qquad V_{ij} = \mathcal{N}\left(0, \frac{1}{n}\right) \tag{46}$$

The expected value of $Q$ is

$$\mathbb{E}[Q] = Var(U_{ij}) \times m \times h + Var(V_{ij}) \times h \times n = m - h. \tag{47}$$

Figure 4 shows the distribution of $Q$ for 2-layer linear NN with different layer dimensions. For each dimension tuples $(m, h, n)$, we constructed 1000 sets of parameters using Xavier initialization. The centers of the distributions of $Q$ match Eq. (47).

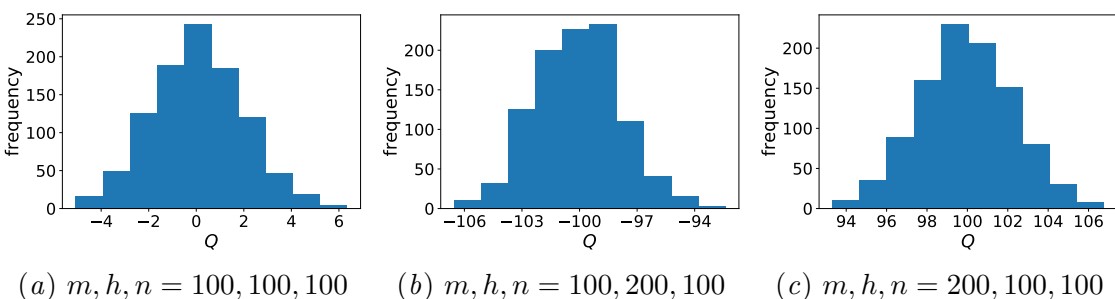

(a) $m, h, n = 100, 100, 100$     (b) $m, h, n = 100, 200, 100$     (c) $m, h, n = 200, 100, 100$

Figure 4: Distribution of $Q$ for 2-layer linear NN with different layer dimensions.

Next, we consider the nonlinear two-layer neural network $U\sigma(VX)$, where $\sigma : \mathbb{R} \to \mathbb{R}$ is an element-wise activation function. For simplicity, we assume whitened input ($X = I$). We choose the following form of the conserved quantity:

$$Q = \frac{1}{2}\operatorname{Tr}[U^T U] - \sum_{a,j} \int_0^{V_{aj}} dx \frac{\sigma(x)}{\sigma'(x)} \tag{48}$$

Figure 5 shows the distribution of $Q$ for 2-layer linear NN with different nonlinearities, each with 1000 sets of parameters created under Xavier initialization. The shapes of the distributions are similar to that of linear networks. The value of $Q$ is usually concentrated around a small range of values. Since the range of $Q$ is unbounded, the Xavier initialization limits the model to a small part of the global minimum.

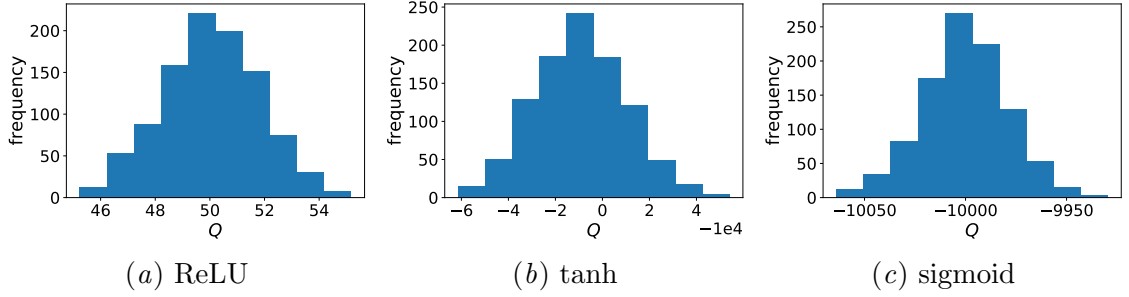

(a) ReLU           (b) tanh           (c) sigmoid

Figure 5: Distribution of $Q$ for 2-layer linear NN with different nonlinearities, with parameter dimensions $m = h = n = 100$.

## Appendix F. Conserved quantity and convergence rate

The values of conserved quantities are unchanged throughout the gradient flow. Since the conserved quantities parameterize trajectories, initializing parameters with certain conserved quantity values accelerates convergence. We derive the relation between conserved quantities and convergence rate for two example optimization problems and provide numerical evidence that initializing parameters with optimal conserved quantity values accelerates convergence.

# Extended Abstract Track

## F.1. Example 1: ellipse

We first show that the convergence rate is related to the conserved quantity in a toy optimization problem. Consider the following loss function with $a \in \mathbb{R}$:

$$\mathcal{L}(w_1, w_2) = w_1^2 + aw_2^2$$
$$\nabla \mathcal{L} = (2w_1, 2aw_2)$$

Assuming gradient flow,

$$\frac{dw_1}{dt} = -\nabla_{w_1} \mathcal{L} = -2w_1 \qquad\qquad \frac{dw_2}{dt} = -\nabla_{w_2} \mathcal{L} = -2aw_2$$

Then $w_1, w_2$ are governed by the following differential equations:

$$w_1(t) = w_{1_0} e^{-2t} \qquad\qquad w_2(t) = w_{2_0} e^{-2at}$$

where $w_{1_0}, w_{2_0}$ are initial values of $w_1$ and $w_2$. We can find conserved quantities by using an ansatz $Q = f(w_1^i w_2^k)$ and solving $\nabla Q \cdot \nabla L = 0$ for $i, k$. Below we use the following form of conserved quantity:

$$Q = \frac{w_1^{2a}}{w_2^2} = \frac{w_{1_0}^{2a}}{w_{2_0}^2}$$

To show the effect of $Q$ on the convergence rate, we fix $L(0)$ and derive how $Q$ affects $L(t)$. Let $L(0) = w_{1_0}^2 + aw_{2_0}^2 = L_0$. Let $w_{2_0}$ continue to be an independent variable. Then $w_{1_0}^2 = L_0 - aw_{2_0}^2$. Substitute in $w_{1_0}^2$, the loss at time $t$ is

$$L(t) = w_1(t)^2 + aw_2(t)^2 = (L_0 - aw_{2_0}^2)e^{-4t} + aw_{2_0}^2 e^{-4at}$$

and $Q$ becomes

$$Q = \frac{w_{1_0}^{2a}}{w_{2_0}^2} = \frac{(L_0 - aw_{2_0}^2)^a}{w_{2_0}^2}$$

The derivative of $L$ in the direction of $Q$ is

$$\partial_Q L(t) = \frac{dL(t)}{dw_{2_0}} \frac{dw_{2_0}}{dQ} = \frac{dL(t)}{dw_{2_0}} \left(\frac{dQ}{dw_{2_0}}\right)^{-1}$$

$$= \left(-2aw_{2_0}e^{-4t} + 2aw_{2_0}e^{-4at}\right) \left(\frac{a(L_0 - aw_{2_0}^2)^{a-1}(-2aw_{2_0})w_{2_0}^2 - 2w_{2_0}(L_0 - aw_{2_0}^2)^a}{w_{2_0}^4}\right)^{-1}$$

$$= \frac{\left(-2aw_{2_0}e^{-4t} + 2aw_{2_0}e^{-4at}\right) w_{2_0}^4}{a(L_0 - aw_{2_0}^2)^{a-1}(-2aw_{2_0})w_{2_0}^2 - 2w_{2_0}(L_0 - aw_{2_0}^2)^a}$$

$$= \frac{2aw_{2_0}^5 \left(e^{-4at} - e^{-4t}\right)}{2w_{2_0}(L_0 - aw_{2_0}^2)^{a-1} \left(-a^2 w_{2_0}^2 - (L_0 - aw_{2_0}^2)\right)}$$

In general, $\partial_Q L(t) \neq 0$, meaning that the loss at time $t$ depends on $Q$. Since we have fixed the initial loss, the convergence rate $L(t) - L(0)$ also depends on $Q$. Special cases where $\partial_Q L(t) = 0$ include $a = 1$ (circle), $a = 0$ (collapsed dimension), and certain initializations such as $w_{2_0} = 0$ (local maximum of gradient magnitude).

### F.2. Example 2: radial activation functions

In this example, we find the conserved quantities and their relation with convergence rate for two-layer reparametrization with radial activation functions under spectral initialization.

Define radial function $g : \mathbb{R}^{m \times n} \to \mathbb{R}^{m \times n}$ as

$$g(W)_{ij} = h\left(|W_i|\right) W_{ij}, \tag{49}$$

where $|W_i| = \left(\sum_k W_{ik}^2\right)^{\frac{1}{2}}$ is the norm of the $i^{th}$ row of $W$, and $h : \mathbb{R} \to \mathbb{R}$ outputs a scalar.

Consider the following objective:

$$\operatorname{argmin}_{U,V}\{\mathcal{L}(U,V) = \frac{1}{2}\|Y - Ug(V^T)\|_F^2\} \tag{50}$$

with spectral initializations

$$U_0 = \Phi\overline{U}_0, V_0 = \Psi\overline{V}_0,$$

where $\Phi, \Psi$ come from the singular value decomposition $Y = \Phi\Sigma_Y\Psi^T$, and $\overline{U}_0, \overline{V}_0$ are random diagonal matrices.

**Proposition 14** *Under the gradient flow $U = -\nabla_U\mathcal{L}$ and $V = -\nabla_V\mathcal{L}$, the following quantity is an invariant:*

$$Q = \frac{1}{2}\operatorname{Tr}[U^TU] - \sum_i \int_{x_0}^{\overline{V}_{ii}} dx \frac{g(x)}{g'(x)} \tag{51}$$

**Proof** Since $g$ is a radial function on rows and $\Psi^T$ is an orthogonal matrix, $g(\overline{V}^T\Psi^T) = g(\overline{V}^T)\Psi^T$. With spectral initialization, the loss function can be reduced to only involving diagonal matrices:

$$
\begin{aligned}
\mathcal{L} &= \frac{1}{2}\|Y - Ug(V^T)\|_F^2 \\
&= \frac{1}{2}\|\Phi\Sigma\Psi^T - \Phi\overline{U}g[(\Psi\overline{V})^T]\|_F^2 \\
&= \frac{1}{2}\|\Phi\Sigma\Psi^T - \Phi\overline{U}g(\overline{V}^T)\Psi^T\|_F^2 \\
&= \frac{1}{2}\|\Phi\left(\Sigma - \overline{U}g(\overline{V}^T)\right)\Psi^T\|_F^2 \\
&= \frac{1}{2}\|\Sigma - \overline{U}g(\overline{V}^T)\|_F^2
\end{aligned}
\tag{52}
$$

Since $\overline{V}$ is a diagonal matrix, $g$ is now an element wise function on $\overline{V}$. Let $\overline{W} = \overline{U}g(\overline{V}^T)$. The gradients for $\overline{U}$ and $\overline{V}$ are

$$
\begin{aligned}
\frac{\partial\mathcal{L}}{\partial\overline{U}} &= \nabla_{\overline{W}}\mathcal{L}g(\overline{V})^T \\
\frac{\partial\mathcal{L}}{\partial\overline{V}} &= \nabla_{\overline{W}}\mathcal{L}^T\overline{U} \odot g'(\overline{V})
\end{aligned}
\tag{53}
$$

where $g'(x) = dg(x)/dx$ is the derivative of the nonlinearity. Additionally, since $\mathcal{L}$ does not depend on $\Phi$ and $\Psi$,

$$\frac{\partial \mathcal{L}}{\partial \Phi} = \frac{\partial \mathcal{L}}{\partial \Psi} = 0 \tag{54}$$

Since the rows of $\Phi, \Psi$ are orthogonal,

$$\frac{\partial \mathcal{L}}{\partial U} = \frac{\partial \mathcal{L}}{\partial \overline{U}} \Phi^T = \nabla_{\overline{W}} \mathcal{L} g(\overline{V})^T \Phi^T$$
$$\frac{\partial \mathcal{L}}{\partial V} = \frac{\partial \mathcal{L}}{\partial \overline{V}} \Psi^T = \left( \nabla_{\overline{W}} \mathcal{L}^T \overline{U} \odot g'(\overline{V}) \right) \Psi^T \tag{55}$$

$\Phi$ and $\Psi$ are not changed in gradient flow, so $\frac{\partial Q}{\partial U} = \frac{\partial Q}{\partial \overline{U}} \Phi^T$ and $\frac{\partial Q}{\partial V} = \frac{\partial Q}{\partial \overline{V}} \Psi^T$. Define inner product on matrices as $\langle X, Y \rangle = \text{Tr}[X^T Y]$. For $Q$ to be a conserved quantity, we need $\langle \nabla \mathcal{L}, \nabla Q \rangle = 0$:

$$\begin{aligned}
\langle \nabla \mathcal{L}, \nabla Q \rangle &= \langle \frac{\partial \mathcal{L}}{\partial U}, \frac{\partial Q}{\partial U} \rangle + \langle \frac{\partial \mathcal{L}}{\partial V}, \frac{\partial Q}{\partial V} \rangle \\
&= \langle \nabla_{\overline{W}} \mathcal{L} g(\overline{V})^T \Phi^T, \frac{\partial Q}{\partial \overline{U}} \Phi^T \rangle + \langle \left( \nabla_{\overline{W}} \mathcal{L}^T \overline{U} \odot g'(\overline{V}) \right) \Psi^T, \frac{\partial Q}{\partial \overline{V}} \Psi^T \rangle \\
&= \langle \nabla_{\overline{W}} \mathcal{L} g(\overline{V})^T, \frac{\partial Q}{\partial \overline{U}} \rangle + \langle \left( \nabla_{\overline{W}} \mathcal{L}^T \overline{U} \odot g'(\overline{V}) \right), \frac{\partial Q}{\partial \overline{V}} \rangle \\
&= \text{Tr} \left[ \partial_{\overline{U}^T} Q \nabla_{\overline{W}} \mathcal{L} g(\overline{V})^T + \overline{U}^T \nabla_{\overline{W}} \mathcal{L} (\partial_V Q \odot g'(\overline{V})) \right] = 0 \tag{56}
\end{aligned}$$

Following the same procedure as for elementwise functions, to have a $Q$ which satisfies (56) it is sufficient to have

$$\frac{\partial Q}{\partial \overline{U}_{ia}} = f(\overline{U}, \overline{V}) \overline{U}_{ia} \qquad \frac{\partial Q}{\partial \overline{V}_{aj}} g'(\overline{V})_{aj} = -f(\overline{U}, \overline{V}) g(\overline{V})_{aj} \qquad f(\overline{U}, \overline{V}) \in \mathbb{R} \tag{57}$$

For simplicity, let $f(\overline{U}, \overline{V}) = 1$. Then, (57) is satisfied by

$$Q = \frac{1}{2} \text{Tr}[\bar{U}^T \bar{U}] - \sum_i \int_{x_0}^{\bar{V}_{ii}} dx \frac{g(x)}{g'(x)} \tag{58}$$

∎

Tarmoun et al. (2021) shows that the conserved quantity $Q$ appears as a term in the convergence rate of the matrix factorization gradient flow. We observe a similar relationship between $Q$ and convergence rate when the loss function is augmented with a radial activation function, as shown in the following proposition.

**Proposition 15** *Consider the objective function and spectral initialization defined in Proposition 14. Let $h\left(|W_i|\right) = |W_i|^{-2}$, and $X = Ug(V^T) = \Phi \Sigma_X \Psi^T$. Then, the eigencomponent of $X$ approaches the corresponding eigencomponent of $Y$ at a rate of*

$$\dot{\sigma}_i^X = \frac{1}{\lambda_i} (\sigma_i^Y - \sigma_i^X)(\sigma_i^{X\,2} + 1)^2, \tag{59}$$

*where $\sigma_i^X = diag(\Sigma_X)_i$, $\sigma_i^Y = diag(\Sigma_Y)_i$, and $\lambda_i = \bar{U}_{ii}^2 + \bar{V}_{ii}^2$ are conserved quantities.*

# Extended Abstract Track

**Proof** Similar to Tarmoun et al. (2021), components can be decoupled, and we have a set of differential equations on scalars:

$$\dot{\overline{u}}_i = [\sigma_i^Y - \overline{u}_i g(\overline{v}_i)]g(\overline{v}_i)$$

$$\dot{\overline{v}}_i = [\sigma_i^Y - \overline{u}_i g(\overline{v}_i)]\overline{u}_i \frac{dg(\overline{v}_i)}{d\overline{v}_i} \tag{60}$$

We also have

$$\dot{g}(\overline{v}_i) = \frac{dg}{d\overline{v}_i}\frac{d\overline{v}_i}{dt} = [\sigma_i^Y - \overline{u}_i g(\overline{v}_i)]\overline{u}_i \left(\frac{dg(\overline{v}_i)}{d\overline{v}_i}\right)^2. \tag{61}$$

Let $\sigma_i^X = \overline{u}_i g(\overline{v}_i)$. Then

$$\dot{\sigma}_i^X = \dot{\overline{u}}_i g(\overline{v}_i) + \overline{u}_i \dot{g}(\overline{v}_i)$$

$$= \left[\sigma_i^Y - \overline{u}_i g(\overline{v}_i)\right]\left[g(\overline{v}_i)^2 + \overline{u}_i^2 \left(\frac{dg(\overline{v}_i)}{d\overline{v}_i}\right)^2\right]. \tag{62}$$

Since $\overline{V}$ is a diagonal matrix, $g$ is now an element wise function on $\overline{V}$. Specifically, $g(\overline{v}_i) = \frac{1}{\overline{v}_i}$. According to Proposition 14, the following quantity is invariant:

$$\frac{1}{2}\overline{u}_i^2 - \int dx \frac{g(x)}{g'(x)} = \frac{1}{2}\overline{u}_i^2 - \int dx \frac{\overline{v}_i^{-1}}{-\overline{v}_i^{-2}} = \frac{1}{2}\overline{u}_i^2 + \frac{1}{2}\overline{v}_i^2 \tag{63}$$

Since any function of the invariant is also invariant, we will use the following form:

$$Q = \overline{U}^T\overline{U} + \overline{V}^T\overline{V}, \tag{64}$$

and define

$$\lambda_i = Q_{ii} = \overline{u}_i^2 + \overline{v}_i^2 \tag{65}$$

Using the $g$ that we defined,

$$\sigma_i^X = \overline{u}_i g(\overline{v}_i) = \overline{u}_i \overline{v}_i^{-1}. \tag{66}$$

In order to relate $\sigma^X$ and $Q$, we first write $\overline{u}_i$ and $\overline{v}_i$ as functions of $\sigma_i^X$ ad $Q$ using (65) and (66):

$$\overline{u}_i^2 = \frac{\lambda_i \sigma_i^{X2}}{\sigma_i^{X2} + 1}, \qquad\qquad \overline{v}_i^2 = \frac{\lambda_i}{\sigma_i^{X2} + 1}. \tag{67}$$

Then, substitute $\bar{u}_i$, $\bar{v}_i$, $g(\bar{v}_i)$, and $\frac{dg(\bar{v}_i)}{d\bar{v}_i}$ into (62), and we have

$$
\begin{aligned}
\dot{\sigma}_i^X &= [\sigma_i - \bar{u}_i g(\bar{v}_i)] \left[ g(\bar{v}_i)^2 + \bar{u}_i^2 \left( \frac{dg(\bar{v}_i)}{d\bar{v}_i} \right)^2 \right] \\
&= [\sigma_i^Y - \bar{u}_i g(\bar{v}_i)] \left[ \left( \frac{1}{\bar{v}_i} \right)^2 + \bar{u}_i^2 \left( -\bar{v}_i^{-2} \right)^2 \right] \\
&= [\sigma_i^Y - \bar{u}_i g(\bar{v}_i)] \left[ (\bar{v}_i^2)^{-1} + \bar{u}_i^2 (\bar{v}_i^2)^{-2} \right] \\
&= [\sigma_i^Y - \sigma_i^X] \left[ \left( \frac{\lambda_i}{\sigma_i^{X2}+1} \right)^{-1} + \frac{\lambda_i \sigma_i^{X2}}{\sigma_i^{X2}+1} \left( \frac{\lambda_i}{\sigma_i^{X2}+1} \right)^{-2} \right] \\
&= [\sigma_i^Y - \sigma_i^X] \left[ \frac{\sigma_i^{X2}+1}{\lambda_i} + \frac{\sigma_i^{X2}(\sigma_i^{X2}+1)}{\lambda_i} \right] \\
&= [\sigma_i^Y - \sigma_i^X] \left[ \frac{\sigma_i^{X4} + 2\sigma_i^{X2} + 1}{\lambda_i} \right] \\
&= \frac{1}{\lambda_i} (\sigma_i^Y - \sigma_i^X)(\sigma_i^{X2} + 1)^2
\end{aligned}
\tag{68}
$$

∎

Proposition 15 relates the rate of change in parameters $\dot{\sigma}_i^X$ and the conserved quantity $\lambda_i$. To get a more explicit expression of how $\lambda_i$ affects convergence rate, we will derive a bound for $|\sigma_i^Y - \sigma_i^X|$, which describes the distance between trainable parameters to their desired value.

**Proposition 16** *The difference between the singular values of $Ug(V^T)$ and $Y$ is bounded by*

$$
|\sigma_i^X - \sigma_i^Y| \le |\sigma_i^X(0) - \sigma_i^Y| e^{-\frac{t}{\lambda_i}}.
\tag{69}
$$

**Proof** Note that

$$
\dot{\sigma}_i^X = \frac{1}{\lambda}(\sigma_i^Y - \sigma_i^X)(\sigma_i^{X2} + 1)^2 \ge \frac{1}{\lambda_i}(\sigma_i^Y - \sigma_i^X)
\tag{70}
$$

Consider the following two differential equations, with same initialization $a(0) = b(0)$:

$$
\begin{aligned}
\dot{a} &= \frac{1}{\lambda}(\sigma - a)(a^2 + 1)^2 \\
\dot{b} &= \frac{1}{\lambda}(\sigma - b)
\end{aligned}
\tag{71}
$$

In these equations, both $a$ and $b$ moves from $a(0) = b(0)$ to $\sigma$ monotonically. Since $\dot{a} \ge \dot{b}$ at every $a = b$, $a$ will always be closer to $\sigma$ than $b$ does. We can explicitly solve for $b$, which yields $b(t) = \sigma + (b(0) - \sigma)e^{-\frac{t}{\lambda}}$. Then the distance between $b$ and $\sigma$ is $|b - \sigma| = |b(0) - \sigma|e^{-\frac{t}{\lambda}}$. Using $|b - \sigma|$, we can bound $|a - \sigma|$:

$$
|a - \sigma| \le |b - \sigma| = |b(0) - \sigma|e^{-\frac{t}{\lambda}}
\tag{72}
$$

Extended Abstract Track

Therefore,

$$|\sigma_i^X - \sigma_i^Y| \le |\sigma_i^X(0) - \sigma_i^Y|e^{-\frac{t}{\lambda_i}} \tag{73}$$

∎

Since $\lambda$ is a conserved quantity, its value set at initialization remains unchanged throughout the gradient flow. Therefore, we are able to optimize the convergence rate by choosing a favorable value for $\lambda$ at initialization. In this example, smaller $\lambda_i$'s lead to faster convergence.

### F.3. Experiments

We compare the convergence rate of two-layer networks initialized with different $Q$ values. We run gradient descent on two-layer networks with whitened input with the following objective

$$\operatorname{argmin}_{U,V}\{\mathcal{L}(U,V) = \|Y - U\sigma(V^T)\|_F^2\} \tag{74}$$

where $\sigma$ is the identity function, ReLU, sigmoid, or tanh. $Y \in \mathbb{R}^{5\times10}$, $U \in \mathbb{R}^{5\times50}$ and $V \in \mathbb{R}^{10\times50}$ have random Gaussian initialization with zero mean. We repeat the gradient descent with learning rate 0.1, 0.01, and 0.001. The learning rate is set to $10^{-3}$, as we do not observe significant changes in the shape of learning curves at smaller learning rates. $U$ and $V$ are initialized with different variance, which leads to different initial values of $Q$. As shown in Fig.6, the number of steps required for the loss curves to drop to near convergence level is correlated with $Q$ in both linear and element-wise nonlinear networks. This result provides empirical evidence that initializing parameters with optimal values for $Q$ accelerates convergence.

We then demonstrate the effect of conserved quantity values on the convergence rate of radial neural networks. Fig.7 shows the training curve for loss function defined in Proposition 15. We initialize parameters $U \in \mathbb{R}^{5\times5}$ and $V \in \mathbb{R}^{10\times5}$ with 4 different values of $Q$ and the learning rate is set to $10^{-5}$. As predicted in Eq. 69, convergence is faster when $Q = \text{Tr}[U^TU + V^TV]$ is small.

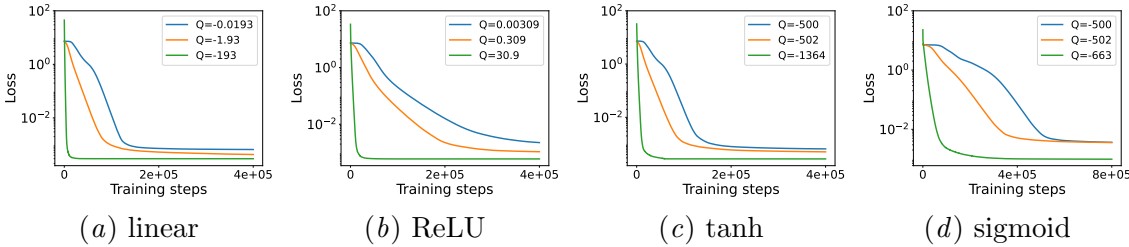

$(a)$ linear $\qquad$ $(b)$ ReLU $\qquad$ $(c)$ tanh $\qquad$ $(d)$ sigmoid

Figure 6: Training curves of two-layer networks initialized with different $Q$. The value of $Q$ affects convergence rate.

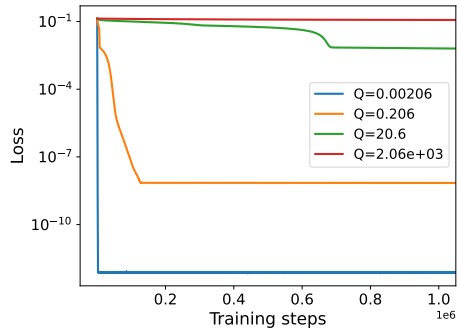

Figure 7: Training curve for the loss function defined in Proposition 15. Smaller value of $Q = \text{Tr}[U^T U + V^T V]$ at initialization leads to faster convergence.

## Appendix G. Conserved quantity and generalization ability

Conserved quantities parameterize the minimum of neural networks and are related to the eigenvalues of the Hessian at minimum. Recent theory and empirical studies suggest that sharp minimum do not generalize well (Hochreiter and Schmidhuber, 1997; Keskar et al., 2017; Petzka et al., 2021). Explicitly searching for flat minimum has been shown to improve generalization bounds and model performance (Chaudhari et al., 2017; Foret et al., 2020; Kim et al., 2022). We derive their relationship for the simplest two-layer network, and show empirically that conserved quantity values affect sharpness. Like convergence rate, a systematic study of the relationship between conserved quantity and generalization ability of the solution is an interesting future direction.

### G.1. Example: two-layer linear network with 1D parameters

We again consider the two-layer linear network with loss $\mathcal{L} = \frac{1}{2}\|Y - UVX\|^2$. For simplicity, we work with one dimensional parameters $U, V \in \mathbb{R}$ and assume $X = Y = 1$ in this example. We show that at the point to which the gradient flow converges, the eigenvalues of the Hessian are related to the value of the conserved quantity.

The gradients and Hessian of $\mathcal{L}$ are

$$\nabla \mathcal{L} = \begin{bmatrix} -(Y - UVX)VX \\ -(Y - UVX)UX \end{bmatrix} \qquad \mathcal{H} = \begin{bmatrix} V^2 X^2 & -YX + 2UVX^2 \\ -YX + 2UVX^2 & U^2 X^2 \end{bmatrix} \qquad (75)$$

At the minima, $U, V$ are related by $UVX = Y$. Recall that $Q = U^2 - V^2$ is a conserved quantity. From the above two equations, we can write $U, V$ as functions of $Q$. Taking the solution $U = \sqrt{\frac{1}{2}(Q + \sqrt{Q^2 + 4})}, V = \sqrt{\frac{1}{2}(-Q + \sqrt{Q^2 + 4})}$ and substitute in $X = Y = 1$, we have

$$\mathcal{H} = \begin{bmatrix} \frac{1}{2}(-Q + \sqrt{Q^2 + 4}) & 1 \\ 1 & \frac{1}{2}(Q + \sqrt{Q^2 + 4}) \end{bmatrix}, \qquad (76)$$

and the eigenvalues of $\mathcal{H}$ are

$$\lambda_1 = 0, \qquad\qquad \lambda_2 = 2\sqrt{Q^2 + 4}. \qquad (77)$$

We have shown that $Q$ is related the eigenvalues of the Hessian at the minimum. Since the eigenvalues determines the curvature, $Q$ also determines the sharpness of the minimum, which is believed to related to model's generalization ability. The result in this example can also be observed in Figure 1, where the minimum of the $Q = 0$ trajectory lies at the least sharp point of the loss valley.

### G.2. Experiments: two-layer networks

The goal of this section is to explore the relation between $Q$ and the sharpness of the trained model. We measure sharpness by the magnitude of the eigenvalues of the Hessian, which are related to the curvature at the minima. We use the same loss function (74) in Section F.3. The parameters are $U \in \mathbb{R}^{10\times50}$ and $V \in \mathbb{R}^{5\times50}$, each initialized with zero mean and various standard deviations that lead to different $Q$'s. We first train the models using gradient descent. We then use the vectorized parameters in the trained model to compute the eigenvalues of the Hessian.

The linear model extends the example in Section G.1 to higher dimension parameter spaces. 700 out of the 750 eigenvalues are around 0 (with magnitude $\leq 10^{-3}$). After removing the small eigenvalues, the center of the eigenvalue distribution correlates positively with the value of $Q$ (Figure 8($a$)). In models with nonlinear activations, $Q$ is still related to eigenvalue distributions, although the relations seem to be more complicated.

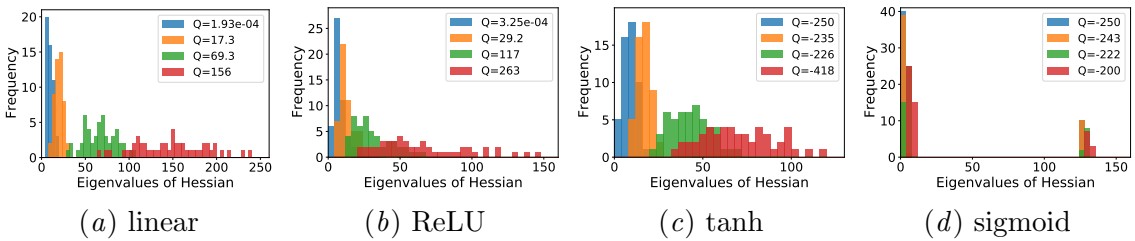

$(a)$ linear $\qquad$ $(b)$ ReLU $\qquad$ $(c)$ tanh $\qquad$ $(d)$ sigmoid

Figure 8: Eigenvalues of the Hessian from trained models initialized with different conserved quantity values ($Q$). The distribution of the eigenvalues and the value of $Q$ appear to be related.

## Appendix H. Ensemble models

In neural networks, the optima of the loss functions are connected by curves or volumes, on which the loss is almost constant (Freeman and Bruna, 2017; Garipov et al., 2018; Draxler et al., 2018; Benton et al., 2021; Izmailov et al., 2018). Various algorithms have been proposed to find these low-cost curves, which provides a low-cost way to create an ensemble of models from a single trained model. Using our group actions, we propose a new way of constructing models with similar loss values. We show that even with stochasticity in the data, the loss is approximately unchanged under the group action (Appendix H). This provides an efficient alternative to build ensemble models, since the transformation only requires random elements in the symmetry group, without any searching or additional optimization.

# Extended Abstract Track

We implement our group actions by modifying the activation function between two consecutive layers. Let $H = VX$ be the output of the previous layer. The group action on the weights $U, V$ is

$$g \cdot (U, V) = (U\pi(g, H), gV) \tag{78}$$

where $\pi(g, H) = \sigma(H)\sigma(gH)^\dagger$. The new activation implements the symmetry group action

$$U\sigma(H) \to U\pi(g, H)\sigma(gH) \tag{79}$$

by wrapping the transformations around an activation function $\sigma'(x) = \pi(g, x)\sigma(gx)$, so that $U\sigma'(H) = U\pi(g, H)\sigma(gH)$.

We test the group action on CIFAR-10. The model contains a convolution layer with kernel size 3, followed by a max pooling, a fully connected layer, a leaky ReLU activation, and another fully connected layer. The group action is on the last two fully connected layers. After training a single model, we create transformed models using $g = I + \varepsilon M$, where $M \in \mathbb{R}^{32 \times 32}$ is a random matrix and $\varepsilon$ controls the magnitude of movement in the parameter space. We then use the mode of the transformed models' prediction as the final output.

We compare the ensemble formed by group actions to four ensembles formed by various random transformation. Let $g = I + \varepsilon M$. The random baselines are:

- 'group': $(U, V) \mapsto (U\pi(g, H), gV)$. This is the model created by group actions.

- '$g^{-1}$': $(U, V) \mapsto (Ug^{-1}, gV)$.

- 'random': $(U, V) \mapsto (Ug', gV)$, where $g' = I + \varepsilon D$ and $D$ is a random diagonal matrix.

- 'shuffle': $(U, V) \mapsto (U\pi'(g, H), gV)$, where $\pi'(g, H)$ is constructed by randomly shuffling $\pi(g, H)$.

- 'interpolated permute' or 'perm_interp': $(U, V) \mapsto (U \left( \frac{I + \frac{\varepsilon}{2}(I+S)}{I+\varepsilon} \right)^{-1}, \frac{I + \frac{\varepsilon}{2}(I+S)}{I+\varepsilon} V)$, where $S \in \mathbb{R}^{32 \times 32}$ is a random permutation matrix.

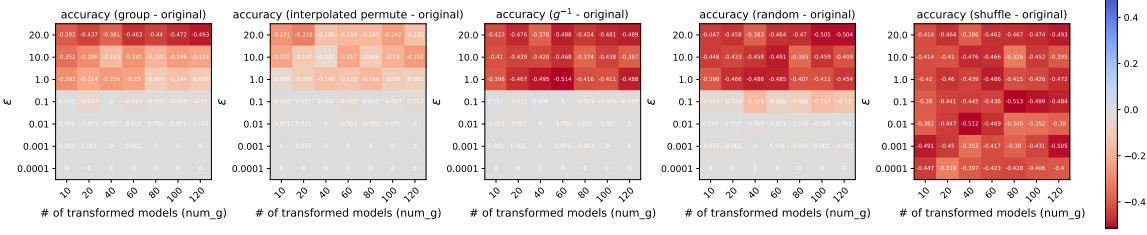

Figure 9: Change in accuracy compared to the original single model when using the ensemble model and 4 baselines. The red color indicates degradation in model performance. The ensemble created by group actions has similar loss values when $\varepsilon$ is small.

Extended Abstract Track

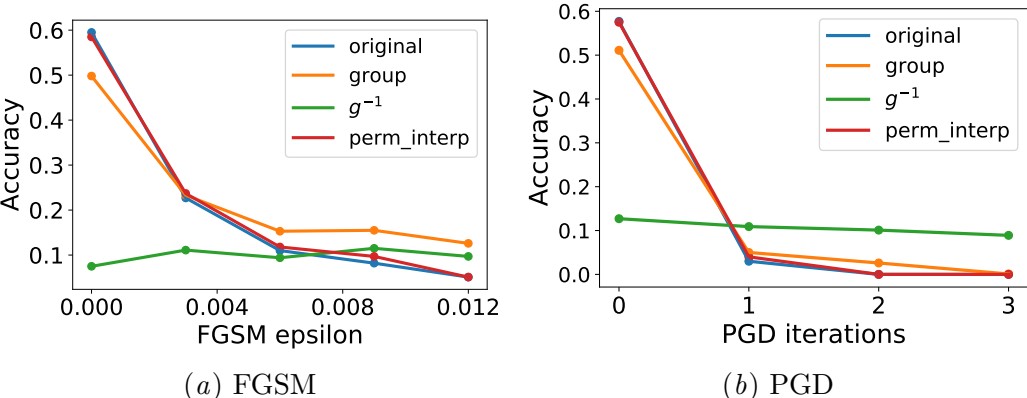

(a) FGSM          (b) PGD

Figure 10: Adversarial attacks on the original model and the ensemble models with various strengths. In FGSM, the group ensemble model improves robustness. In PGD, the ensemble has negligible effects on robustness.

Figure 9 shows the accuracy of the ensembles compared to single models. The ensemble formed by group actions preserves the model accuracy for small $\varepsilon$ and has smaller accuracy drop at larger $\varepsilon$. The ensemble model also improves robustness against Fast Gradient Signed Method (FGSM) attacks (Figure 10). Under FGSM attacks with various strength, the ensemble model created using group actions consistently performs better than the baselines with random transformations. However, the same improvement is not observed under Projected Gradient Descent (PGD) attacks.

