# OpenReview forum: "Charting Flat Minima Using the Conserved Quantities of Gradient Flow"
_NeurIPS.cc/2022/Workshop/NeurReps — NeurReps 2022 Poster_

### Official Review · Reviewer_L4bg · 2022-10-10
**A new insight on flat minima with solid analysis**

**Confidence:** 3
**Soundness:** 3
**Presentation:** 3
**Contribution:** 3
**Overall Rating:** 7

**Summary:**

The paper proposes a characterization of a low-loss valley through conserved quantities of the gradient flow. Firstly they define and characterize continuous symmetries of the loss as action on the parameter space. Then they derive a definition of conserved quantities of gradient flow based on this action. Their formulation of conserved quantities is almost stable to time discretization, making it usable on gradient descent.
Since the conserved quantities are determined at initialization and unchanged during gradient flow, they can parametrize the gradient flow trajectories. They demonstrate how conserved quantities influence the convergence rate and the generalization ability through preliminary results. Finally, they propose a new way of constructing ensemble models based on symmetry groups.

**Questions:**

Some parts are not clear to me. After Proposition 3, it is unclear what $Q_{M_s}$ stands for. Why is it equal to $Q_M$?
In a few lines below, the authors refer to "GD". Does it stand for gradient descent? It would be better to introduce this notation.

I wonder how feasible and expensive it would be to extend the paper's analysis to neural networks composed of more than 2 layers. Does the computation of Q become more complex? For instance, does the performance of the ensemble model shown in Figure 7 become more sensitive to $\epsilon$?

Given Proposition 3, how does the choice of M affects the conserved quantity Q? Does different M lead to a distinct analysis of the parameters space?


**Limitations:**

A potential limitation is how this analysis is affected by the size of the neural network and the choice of M.


**Recommended Decision:**

3: Accept

**Relevance:**

4: Highly relevant

**Strengths And Weaknesses:**

The novelty of the work is the definition of conserved quantities through action as symmetries of the loss. Their formulation allows for characterizing a model's convergence rate and generability in a new way. Moreover, they suggest how to construct ensemble models based on symmetry groups. I find this an exciting improvement.

The propositions are supported with demonstrations and practical examples. Moreover, future directions are supported by preliminary results, resulting in a solid direction.

The paper is well organized but challenging to follow in some parts. See Question for more details.

The leverage of action as symmetries of the loss is an exciting approach that fits this community. This work can bring new theoretical insight in addition to promising applications.


**Submission Track:**

Extended Abstract (4 Page)

---

### Official Review · Reviewer_KGDn · 2022-10-14
**Unclear what the contribution is**

**Confidence:** 3
**Soundness:** 2
**Presentation:** 1
**Contribution:** 2
**Overall Rating:** 4

**Summary:**

The paper explores the idea that "low-loss valleys" are connected to invariances of the loss under certain lie group symmetries and how this affects parameter estimation.

**Questions:**

see above.

**Limitations:**

see above

**Recommended Decision:**

2: Borderline

**Relevance:**

3: Solid fit

**Strengths And Weaknesses:**

The basic idea is quite attractive: if a loss is invariant under a lie group action on the parameters then it means that along those paths / or manifolds the loss will be constant and this has multiple interesting consequences for gradient descent that should be carefully explored.

The extended abstract itself explains this in mostly abstract terms. The results summarized here are mathematically speaking entirely obvious but are a bit blown out of proportion by stating them as formal lemmas, etc. Everything in section 2 and 3 (except Prop 2 - more on this below) can be compressed into about 1/2 page. Section 4 (applications and future) work is where we finally get to the consequences of those elementary observations and those are indeed intersting but unfortunately no further details are given. I'm afraid I do not have the time to dig through 25 pages or appendices. Please summarize your most intersting results in the main text.

Regarding Prop 2: It looks like a potentially interesting result, but I cannot check it because you didn't specify your terms or discussed your assumptions. What is U, V, sigma? and why is the full-rank assumption justified?

Finally - the presentation as is makes it very unclear what is new and what is summary. This must be more clearly stated. E.g. I have no idea whether Prop 2 is new? It would be remarkable if it is.

In summary - I think this abstract discusses an interesting topic but the quality of the submission (as is) is not appropriate for this workshop.

**Submission Track:**

Extended Abstract (4 Page)

---

### Official Review · Reviewer_PUJm · 2022-10-17
**Elegant connection of parameter symmetries to flat minima**

**Confidence:** 3
**Soundness:** 3
**Presentation:** 3
**Contribution:** 4
**Overall Rating:** 7

**Summary:**

Modern deep neural networks are overparameterized and frequently exhibit minima with flat directions. This paper provides theoretical insight into the nature of these flat minima by connecting them to the symmetries of the parameterization with respect to the loss function. Expressing these symmetries through the actions of Lie groups on the network parameters, the authors show that the infinitesimal generators of such groups define flat directions. They further show how these infinitesimal generators can be used to derive invariants of the gradient flow dynamics, which thus parameterize the dynamics. Such parameterization is important because as the authors show analytically in simple examples and numerically in more realistic examples, convergence time to minima and the curvature of the minima found can depend strongly on these invariants,  and at least one existing initializiation scheme (Xavier) explores only a limited set of these minima, with potentially adverse affects on the solutions found.

**Questions:**

- Are all flat directions due to symmetries in the parameterization? How would a practitioner distinguish those that are from those that aren't? And how would one derive the symmetries from the flat directions? Perhaps by starting with verifying the conditions for a Lie algebra (existence of the bracket, etc.)?

- The authors show numerically the intriguing dependence of convergence dynamics on the invariants derived from the symmetries. Can the authors suggest how practitioners can use these invariants to optimize convergence dynamics of their networks, beyond ensuring that their initializations sample these invariants broadly (and thus hopefully find good dynamics by chance)?

- How difficult is it in practice to determine the symmetries for a given parameterization? The authors provide examples of how this can be done for some shallow nonlinear networks, but it would have been useful for the authors to determine the symmetries of some realistic networks from the literature.

**Limitations:**

- The approach the authors outline goes derives flat directions from symmetries. However, practitioners are usually faced with flat directions first, and the authors don't specify how to derive symmetries from these, and thus capture many flat directions all at once. One must then resort to trial-and-error to discover the symmetries of one's parameterization. However, the authors to their credit do provide some examples of linear and nonlinear symmetries, which should help in this process.

- The authors show that the symmetries result in flat directions, but do not comment on whether all flat directions are the results of symmetries. Thus it's unclear to what exent their approach accounts for the flat minima practitioners have found.

- To their credit, the authors provide many examples of their approach in the main text and the appendices. Many of these are toy examples, which are very useful for pedagological purposes. However, there are only a few examples where the authors apply their approach to previous work so it was sometimes unclear to what extent their approach can be applied in practice to the models they cite from the literature.

**Recommended Decision:**

3: Accept

**Relevance:**

4: Highly relevant

**Strengths And Weaknesses:**

The connection of symmetries to the flat directions and conserved quantities is elegant and should be a very useful tool for practitioners studying the loss landscapes and convergence dynamics of deep neural networks. The appendices are full of examples applying the approach to toy networks which helped build intuition for the approach, though I would have appreciated a few more examples of the approach to the models in the literature to really demonstrate its effectiveness. I worked through the main text and most of the appendices and the work appears to me to be sound. However the exposition was a bit short - the main text providing a very high-level outline of the approach and shifted the bulk of the content to the appendices, which made the main results harder to piece together. This submission might have been more appropriate as a Proceedings Paper. Nevertheless, the results here are important and should be of significant interest to the community. I personally really enjoyed working through this paper and learned a lot.

**Submission Track:**

Extended Abstract (4 Page)

---

### Decision · Program_Chairs · 2022-10-21

Accept (Poster)